**SPECIAL ISSUE**
**CILIA AND FLAGELLA: FROM BASIC BIOLOGY TO DISEASE**

# Prostaglandin E2 inhibits adipogenesis through the cilia-dependent activation of ROCK2

Mark D. Lee and Keren I. Hilgendorf*

## ABSTRACT

Functional adipose tissue is essential for maintaining systemic metabolic homeostasis. Dysfunctional adipose tissue, characterized by increased fibrosis, hypoxia and chronic inflammation, is often associated with obesity and promotes the onset of metabolic disease, such as type 2 diabetes. During nutrient excess, adipose tissue function can be preserved by the generation of new adipocytes from adipocyte stem cells, illustrating the importance of identifying the physiological regulators of adipogenesis. Here, we discover a cilia-localized signaling pathway through which the pro-inflammatory lipid metabolite prostaglandin E2 (PGE2) suppresses adipogenesis. We demonstrate that PGE2 specifically signals through the E-type prostaglandin receptor 4 (EP4) localized to the primary cilium of adipocyte stem cells. Activation of ciliary EP4 initiates a cAMP-independent signaling cascade that activates Rho-associated protein kinase 2 (ROCK2), resulting in the retention of actin stress fibers that prevent adipogenesis. These findings uncover a compartmentalized regulatory mechanism of adipogenesis by which primary cilia alter whole-cell physiology, cell fate, and ultimately adipose tissue expansion in response to an inflammatory hormone, offering insight into how chronic inflammation may contribute to adipose tissue dysfunction and metabolic disease progression.

KEY WORDS: Primary cilia, Adipogenesis, PGE2, Inflammation, ROCK2, Cytoskeleton

## INTRODUCTION

The global rise in obesity over the past several decades has made it a leading contributor to mortality, largely due to its association with co-morbidities, such as type 2 diabetes, cardiovascular disease, and 13 different types of cancer (Janić et al., 2025). In the USA alone, annual healthcare costs related to obesity exceed $200 billion (Cawley et al., 2021), reflecting a profound societal and economic burden. However, metabolic disease is not an inevitable consequence of obesity; instead, metabolic disease is strongly linked to white adipose tissue dysfunction independent of total fat mass (Klöting et al., 2010).

Department of Biochemistry, University of Utah School of Medicine, Salt Lake City, UT 84112, USA.

*Author for correspondence (keren.hilgendorf@biochem.utah.edu)

M.D.L., 0000-0002-9083-2323; K.I.H., 0000-0001-8377-8384

White adipose tissue is highly dynamic and expands with excess nutrient intake. This expansion requires coordinated hypertrophic enlargement of existing adipocytes and the differentiation of nascent adipocytes from resident adipocyte stem cells (ASCs). This process of differentiation is known as adipogenesis and is characterized by numerous gene expression changes allowing for lipid accumulation in the differentiating ASC, ultimately culminating in the formation of a unilocular lipid droplet in a rounded mature adipocyte (Scamfer et al., 2022).

Excessive adipocyte hypertrophy leads to tissue hypoxia, inflammation and fibrosis, hallmarks of dysfunctional adipose tissue (Sun et al., 2011). While adipose tissue dysfunction is associated with obesity, some individuals with obesity maintain metabolic health, displaying reduced inflammation and smaller adipocytes within visceral white adipose tissue (Vishvanath and Gupta, 2019). These observations support a model in which promoting adipogenesis to generate a more numerable pool of smaller adipocytes may preserve adipose tissue function and mitigate the metabolic consequences of obesity. Consistent with this, a class of drugs known as glitazones has been used clinically to improve metabolic health by directly activating the master regulator of adipogenesis, peroxisome proliferator-activated receptor gamma (PPARγ), to promote adipogenesis (Crossno et al., 2006; Czoski-Murray et al., 2004; Shao et al., 2018). There is significant interest in identifying the physiological factors that hinder adipogenesis to better understand the drivers of obesity-linked adipose tissue dysfunction.

ASC differentiation is influenced by numerable intrinsic and extrinsic factors. Recent studies suggest that several of these factors converge on the primary cilium – a microtubule-based signaling organelle present on most mammalian cells, including ASCs – to regulate adipogenesis. Disorders that affect the function of these primary cilia, such as Biedl–Bardet Syndrome and Alström Syndrome, are associated with excessive adiposity and early onset type 2 diabetes (Zhang et al., 2024), underscoring the importance of primary cilia function in metabolic health regulation. Primary cilia serve as specialized cellular compartments enriched in specific signaling components. Although the ciliary membrane is contiguous with the rest of the cell, its contents must be specifically trafficked into and out of the cilium, resulting in pools of compartmentalized signal transducers discrete from the cell body (Hilgendorf et al., 2024). The primary cilium is nucleated by the centrosome, allowing ciliary signal transduction to have privileged access to the central signaling hub of the cell, contextualizing why primary cilia are implicated in cell fate regulation across diverse tissues, including adipose tissue (Hilgendorf et al., 2024; Stoufflet et al., 2020).

A growing number of G protein-coupled receptors (GPCRs) have been found to be selectively trafficked to primary cilia (Hansen et al., 2025; Hilgendorf et al., 2024; Wachten and Mick, 2021).

GPCRs represent the most widely targeted receptor class by FDA-approved drugs (Sriram and Insel, 2018), and there is growing interest in elucidating the therapeutic role of GPCRs in the context of primary cilia (Collinson and Tanos, 2025; Gradilone et al., 2017; Zhu et al., 2025). Our lab previously demonstrated that ciliary GPCRs regulate ASC adipogenic fate, as ASCs lacking ciliary GPCRs exhibit attenuated adipogenesis both *in vitro* and *in vivo* (Hilgendorf et al., 2019), although few studies have investigated which GPCRs localize to ASC cilia. One receptor, free fatty acid receptor 4 (FFAR4), has been shown to increase adipogenesis in a cilia-dependent manner (Hilgendorf et al., 2019), whereas Hedgehog signaling inhibits adipogenesis through ciliary signaling (Scamfer et al., 2022; Spinella-Jaegle et al., 2001). These data suggest primary cilia may act as an adipogenic rheostat, interpolating pro- and anti-adipogenic signals to selectively control the activation of adipogenic transcriptional programs.

The prevalence of extrinsic signals, such as GPCR ligands, can be dramatically altered in dysfunctional adipose tissue (Civelek and Ozen, 2022). Among these are prostaglandins, lipid-derived hormones that directly engage with GPCRs to regulate inflammation, lipolysis, vasodilation, immune cell recruitment, fibrosis and adipogenesis (Bowery et al., 1970; Hu et al., 2016; Xu et al., 2016). As prostaglandin synthesis is upregulated during inflammation (Ricciotti and FitzGerald, 2011), these hormones may play crucial roles in orchestrating metabolic dysfunction during obese adipose tissue expansion. Here, we sought to investigate the mechanism through which the most abundant prostaglandin, PGE2, regulates ASC cell fate.

We demonstrate that PGE2 inhibits adipogenesis specifically via the GPCR prostaglandin E receptor 4 (EP4). We show that EP4 localizes to ASC primary cilia and that this compartmentalization is necessary for its inhibitory effect on adipogenesis. Ciliary EP4 does not inhibit adipogenesis through canonical cAMP signaling in ASCs, but rather through activation of Rho-associated protein kinase 2 (ROCK2). ROCK2 activity is sufficient to inhibit adipogenesis by preventing the remodeling of the ASC actin cytoskeleton, which is required to accommodate lipid droplet formation during adipogenesis. Together, these data highlight a regulatory pathway in adipogenesis that relies on compartmentalized signaling within primary cilia to guide adipose tissue expansion and ultimately influence metabolic health.

## RESULTS
### PGE2 inhibits adipogenesis through the receptor EP4
PGE2 is elevated in obese white adipose tissue (García-Alonso et al., 2016), and both murine and human ASCs from obese individuals express PGE2 receptors, EP1-4 (PTGER1-4), at higher levels than their lean counterparts (Fig. S1A,B) (Emont et al., 2022). Previous studies have demonstrated that PGE2 can remodel adipose tissue by altering mature adipocyte physiology or modulating ASC differentiation (García-Alonso et al., 2016; Vassaux et al., 1992; Wang et al., 2022), so we sought to first identify the time window during which PGE2 affects adipocyte differentiation. 3T3-L1 preadipocytes were grown to confluency, and adipogenesis was initiated using an IDX differentiation cocktail composed of insulin, dexamethasone and IBMX, as previously described (Hwang et al., 1997) (Fig. 1A). PGE2 or vehicle control was added to the cells during various intervals prior to or during adipogenesis, and intracellular lipid content was measured throughout differentiation by live cell imaging using the fluorescent lipophilic dye BODIPY 493/503 (Fig. S1C,D). PGE2 reduced lipid accumulation by nearly 60% in 3T3-L1 cells when present during the first 96 h of

adipogenesis (Fig. 1B), and reduced the size and frequency of intracellular lipid droplets (Fig. 1C); addition of PGE2 prior to IDX treatment or after this critical 96-h window had no effect on 3T3-L1 adipogenesis (Fig. S1D).

To confirm that PGE2 inhibits the commitment to adipogenesis, we assessed its effect on the expression of key adipogenic genes in differentiating 3T3-L1 preadipocytes. PGE2 attenuated the expression of the gene encoding the master regulator of adipogenesis Pparγ and its target genes *Cebpa*, *Adipoq* and *Fabp4* (Fig. 1D) (Lefterova et al., 2008). We validated the inhibitory effect of PGE2 using primary murine ASCs isolated from white adipose tissues by fluorescence activated cell sorting (FACS) as previously reported (Rodeheffer et al., 2008) (Fig. S1E). PGE2 completely abrogated the differentiation of ASCs isolated from the inguinal white adipose depot of both male and female lean mice (Fig. 1E,F, Fig. S1F,G). Similarly, PGE2 inhibited adipogenesis in male perigonadal ASCs (Fig. S1F,H) but was less effective at inhibiting female perigonadal ASC differentiation (Fig. S1G,I), either due to sex-linked differences in expression of the PGE2 receptors (Fig. S1A) or lower overall *ex vivo* adipogenesis rates. Together, these data demonstrate robust inhibition of adipogenesis by PGE2 *in vitro* and *ex vivo*.

We next sought to explore how PGE2 inhibits adipogenesis. PGE2 signals via four GPCRs, EP1-4, all of which are expressed in 3T3-L1 cells (Fig. S1J); to identify which of these receptors is required for the anti-adipogenic effect of PGE2, 3T3-L1 cells were treated with PGE2 in the presence of individual EP receptor antagonists. Notably, only EP4 antagonism was sufficient to rescue adipogenesis in the presence of PGE2, while the activity of EP1, EP2 or EP3 was not required for PGE2 activity in this context (Fig. 1G). To confirm that PGE2 inhibits adipogenesis through EP4, we generated 3T3-L1 preadipocytes lacking EP4 using Crispr/Cas9. Knockout efficiency was evaluated with TIDE sequence analysis (Brinkman et al., 2014) (Fig. S1K) and immunoblotting (Fig. S1L), confirming robust EP4 depletion in this heterogenous EP4 knockout pool. Unlike control 3T3-L1 cells, these EP4 knockouts were not sensitive to PGE2 (Fig. 1H,I). Finally, an EP4-specific agonist recapitulated the effect of PGE2 on adipogenesis in 3T3-L1 cells (Fig. 1J). Thus, PGE2 inhibits adipogenesis specifically through the receptor EP4.

### EP4 localizes to the primary cilium of ASCs
Previous studies have demonstrated that EP4 is enriched within primary cilia of retinal epithelial cells, kidney ductal cells and pancreatic islet cells (Hansen et al., 2022; Jin et al., 2014; Wu et al., 2021), although the subcellular localization of EP4 in ASCs has not previously been explored. ASCs are near uniformly ciliated, although these cilia are lost during adipogenesis and are not present on mature adipocytes (Forcioli-Conti et al., 2015; Hilgendorf et al., 2019; Marion et al., 2009). This cilia loss occurred 96 h post-treatment with the IDX cocktail in 3T3-L1 cells, coincident with when these cells lost their sensitivity to PGE2 (Fig. S1D). We hypothesized that EP4 is also trafficked to ASC primary cilia, and that these cilia regulate PGE2 and EP4 signal transduction during adipogenesis.

To assess ciliary localization of EP4 in ASCs, confluency-arrested 3T3-L1 cells were immunostained for EP4, the centrosome marker FGFR1OP, and the ciliary GTPase ARL13B. Consistent with prior reports (Hilgendorf et al., 2019), 80% of confluency-arrested 3T3-L1 preadipocytes were ciliated and we observed ciliary enrichment of EP4 in 45% of all cells, with 60% of all ciliated cells displaying EP4-positive cilia (Fig. 2A, Fig. S2A). EP4 enrichment

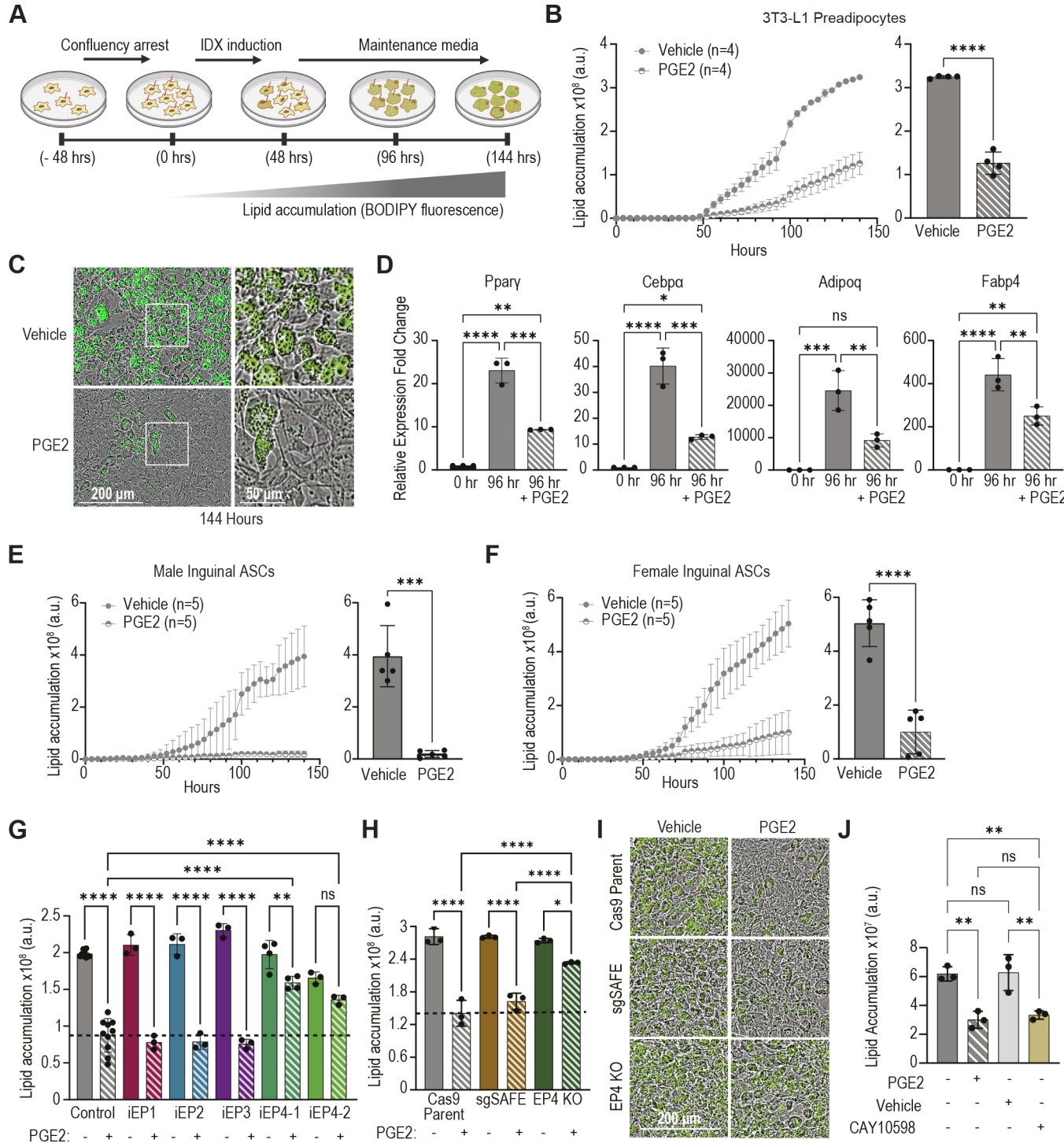

**Fig. 1.** See next page for legend.

in cilia was confirmed in primary ASCs isolated from inguinal white adipose depots from both male and female mice (Fig. 2B,C). While the overall fraction of ciliated cells was reduced in primary ASCs, approximately 60% of all ASC cilia were EP4 positive (Fig. S2A). The EP4 antibody was validated using the 3T3-L1 EP4 knockout cell line (Fig. S2A,B); interestingly, overall ciliation was marginally reduced in the knockout cells compared to controls, consistent with previous reports implicating the PGE2-EP4 signaling axis in ciliogenesis (Jin et al., 2014).

Finally, to confirm that EP4 localization within primary cilia *in vitro* and *ex vivo* was not limited to a 2D culture system, we performed EP4 and cilia immunostaining on whole-mount murine white adipose tissue (Fig. 2D, Fig. S2C,D). Ciliated ASCs were seen situated along the vasculature within inguinal and perigonadal white adipose tissue, as previously described (Hilgendorf et al., 2019; Tang et al., 2008), and EP4 colocalized with ASC cilia *in vivo* in both inguinal (Fig. 2D) and perigonadal (Fig. S2C,D) white adipose tissue.

**Fig. 1. PGE2 inhibits adipogenesis through the G protein-coupled receptor EP4.** (A) Schematic of IDX-induced adipogenesis in ASCs; ciliated cells begin accumulating lipids after the first 48 h of IDX treatment, and lose their cilia near the 96-h mark. Created in BioRender by Lee, M., 2025. https://BioRender.com/m4r2u0d. This figure was sublicensed under CC-BY 4.0 terms. (B) Intracellular lipid content of 3T3-L1 cells after initiation of adipogenesis using the IDX cocktail (t=0 h) in the presence of 20 µM PGE2 or DMSO vehicle during the critical first 96 h of adipogenesis. Lipid content was quantified using the integrated fluorescence intensity of BODIPY 493/503 throughout adipogenesis (left) and at the endpoint (t=144 h) (right). n=number of independent experiments. (C) Representative images showing lipid droplets in differentiated 3T3-L1 preadipocytes (t=144 h) visualized by green fluorescent BODIPY staining. Boxed areas are shown at higher magnification on the right. (D) Relative change in mRNA expression of adipogenic target genes in differentiating 3T3-L1 cells in the presence of 20 µM PGE2 or vehicle control from three independent experiments demonstrates that PGE2 inhibits adipogenic gene expression. (E,F) ASCs isolated from inguinal white adipose depots of male (E) and female (F) mice are sensitive to 20 µM PGE2 as determined by intracellular lipid content throughout adipogenesis (left) and at the endpoint (right). n=number of independent ASC isolations which pooled 2-4 mice each. (G) 3T3-L1 cells treated with the selective antagonists of EP1 (SC-19220, 10 µM), EP2 (PF-04418948, 10 µM), EP3 (L-798,106, 5 µM) and EP4 [MF498 (iEP4-1), 25 µM; MF766 (iEP4-2), 50 µM] 24 h prior to initiation of adipogenesis and then concomitant with 20 µM PGE2 for the first 48 h of adipogenesis. Dashed line marks average lipid content in control cells treated with PGE2 at endpoint. EP4 activity is required for PGE2 to inhibit adipogenesis. (H) Cas9-expressing 3T3-L1 cells, control 3T3-L1 cells expressing a safe sgRNA (sgSAFE) and EP4 knockout cells were treated with 20 µM PGE2 for the first 96 h of adipogenesis. EP4 is required for PGE2 to inhibit adipogenesis. Dashed line marks average endpoint lipid content in Cas9 control cells treated with PGE2. (I) Representative images depicting lipid droplets in cells from the data shown in H. (J) Inhibition of adipogenesis by PGE2 is recapitulated by 48 h treatment with the EP4 agonist CAY10598 (50 µM). (B-H,J) All data are mean±s.d., each data point shows an independent experiment. *P<0.05, **P<0.01, ***P<0.001, ****P<0.0001 [unpaired, two-tailed Student's t-test (B,E,F) or one-way ANOVA followed by Tukey's multiple comparison test (D,G,H,J)]. a.u., arbitrary units; KO, knockout; ns, not significant.

## EP4 localization to primary cilia is required for PGE2 to inhibit adipogenesis

Given that EP4 is highly enriched in the ciliary compartment of ASCs, we hypothesized that this localization is necessary for inhibition of adipogenesis by PGE2. To test this, we depleted kinesin family member 3A (KIF3A), a motor protein required for ciliogenesis, in 3T3-L1 preadipocytes using Crispr/Cas9. Knockout efficiency was assessed by TIDE sequence analysis (Brinkman et al., 2014) (Fig. S3A) with loss of KIF3A protein confirmed by immunoblotting (Fig. S3B). As expected, KIF3A depletion resulted in near-complete loss of primary cilia in 3T3-L1 preadipocytes (Fig. S3C,D). Strikingly, PGE2 failed to inhibit adipogenesis in these knockout cells (Fig. 3A), demonstrating that the primary cilium is necessary for the anti-adipogenic activity of PGE2.

As cilia loss can alter adipogenesis (Zhu et al., 2009), we tested whether blocking entry of EP4 into the cilium would ablate sensitivity to PGE2. Tubby-like protein 3 (TULP3) is an adaptor protein essential for trafficking GPCRs into primary cilia, including EP4 in pancreatic islet cells (Wu et al., 2021). We observed that the loss of TULP3 similarly reduced EP4 localization in primary cilia of 3T3-L1 preadipocytes, resulting in its pericentriolar accumulation at the base of the cilium (Fig. S3E,F). Similar to cilia loss in the KIF3A knockouts, loss of ciliary EP4 localization in TULP3 knockouts eliminated their sensitivity to PGE2 (Fig. 3B). In fact, the predicted EC50 of PGE2 in the TULP3 knockout cells was more than six times greater than in the control 3T3-L1 cells (Fig. 3C).

Next, we tested whether PGE2 could inhibit the physiological signals that induce adipogenesis. The IDX cocktail used to induce adipogenesis in both the KIF3A and TULP3 knockouts contains supraphysiological levels of adipogenesis inducers, which can trigger adipogenesis in a cilia-independent manner. An attenuated version of this cocktail supplemented with the FFAR4 agonist docosahexaenoic acid (DHA) has been shown to induce adipogenesis in a cilia-dependent manner (Hilgendorf et al., 2019). 3T3-L1 cells robustly underwent adipogenesis in response to DHA, and PGE2 cotreatment ablated differentiation induced by this cilia-dependent cocktail (Fig. 3D). Together, these data show that PGE2 signaling is mediated through primary cilia and that the ciliary localization of EP4 is essential for PGE2 to inhibit adipogenesis.

## EP4 does not inhibit adipogenesis through ciliary cAMP

We next explored how the activation of ciliary EP4 by PGE2 may impinge on adipogenesis. Previous studies have shown that activation of ciliary EP4 stimulates cAMP production by coupling to the $G_{\alpha s}$ transducer, resulting in ciliary elongation (Jin et al., 2014). However, other studies have highlighted that EP4 promiscuously couples to other transducers, including $G_{\alpha i}$, $G_{\alpha 12/13}$ and $G_{\alpha 15}$ (Huang et al., 2023; Masuho et al., 2023). To understand how EP4 coupling may alter ciliary dynamics during adipogenesis, we first assessed the effect of PGE2 on cilia length. Consistent with previous findings (Dalbay et al., 2015; Forcioli-Conti et al., 2015), cilia length increased 24 h post-initiation of adipogenesis with the IDX cocktail (Fig. 4A, Fig. S4A,B), and addition of PGE2 further increased cilia length (Fig. 4A, Fig. S4A,B), suggesting that the ability of PGE2 to increase cilia length is conserved across cell types.

Changes in cilia length have been ascribed to increased levels of ciliary cAMP (Hansen et al., 2020). Using a ratiometric fluorescent cAMP biosensor that localizes exclusively to the primary cilium (Moore et al., 2016), we assessed the effect of EP4 activation on cAMP production in 3T3-L1 primary cilia. Changes in cAMP levels were recorded following treatments with the adenylyl cyclase agonist forskolin, the FFAR4 agonist TUG891, PGE2 or a DMSO vehicle control (Fig. 4B). As expected, the sensor measured a sharp, sustained increase in ciliary cAMP levels following forskolin treatment, and activation of the endogenous, cilia-localized, $G_{\alpha s}$-coupled receptor FFAR4 resulted in a comparable increase in ciliary cAMP. Surprisingly, PGE2 treatment did not result in a significant prolonged increase in ciliary cAMP (Fig. 4B); instead, PGE2 treatment led to a minor initial increase of ciliary cAMP that rapidly returned to baseline in the continued presence of PGE2. Thus, the PGE2 stimulus sufficient to inhibit adipogenesis does not exhibit sustained production of ciliary cAMP in 3T3-L1 primary cilia.

To determine whether the observed transient elevation of ciliary cAMP immediately following PGE2 treatment mediates inhibition of adipogenesis by PGE2 we interrogated the role of downstream effectors of cAMP signaling during 3T3-L1 differentiation. Production of cAMP activates two downstream effector proteins, protein kinase A (PKA) and exchange protein activated by cAMP (EPAC; also known as RAPGEF3), and we first tested whether activation of either is sufficient to inhibit adipogenesis. Selective activation of either PKA or EPAC did not recapitulate the effect of PGE2, nor did they synergize with PGE2 when co-treated (Fig. S4C). We next examined whether either effector is required for PGE2 activity and if their inhibition can rescue its effect on adipogenesis. Previous reports have shown EPAC is required for adipogenesis while PKA is dispensable (Ji et al., 2010). Consistent with these observations, treatment with an EPAC-specific inhibitor attenuated adipogenesis, and its co-treatment with PGE2 further suppressed adipogenesis, suggesting that EPAC does not mediate PGE2 activity (Fig. S4C). Inhibition of PKA by itself had no effect

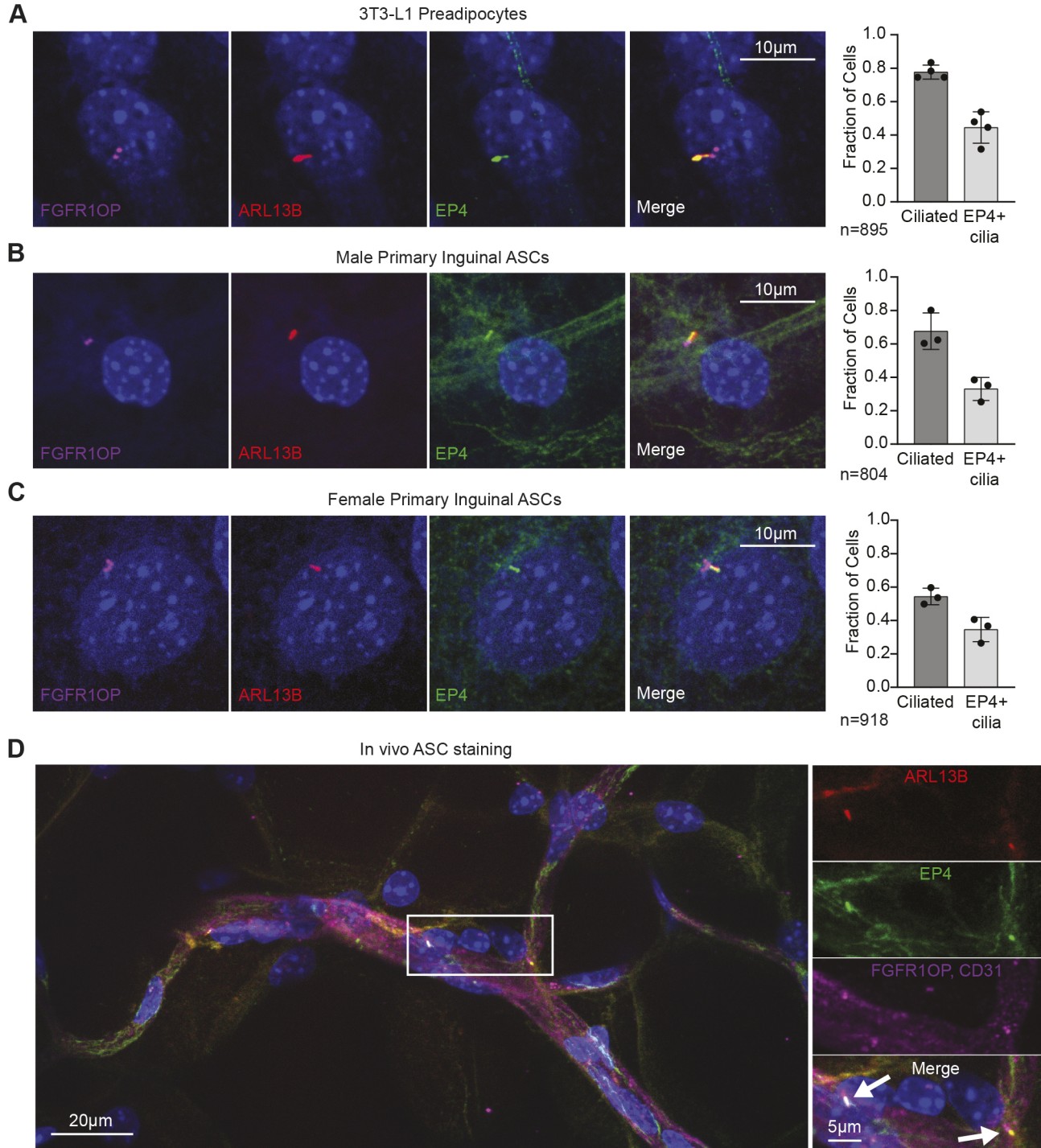

**Fig. 2. EP4 localizes to primary cilia of ASCs.** (A-C) Immunofluorescence staining using DAPI (blue) and antibodies recognizing the centrosomal marker FGFR1OP (magenta), the cilia marker ARL13B (red) and the EP4 receptor (green) in undifferentiated 3T3-L1 cells (A) and primary ASCs from the inguinal fat depot of male (B) and female (C) mice. Left: Representative images showing ciliated cells with enrichment of EP4 in the primary cilium. Right: The fraction of cells with primary cilia and the fraction of cells with EP4-positive cilia; $n$= total cell number. All cells have a nucleus and are positive for the centrosomal marker FGFR1OP, while a subset of cells have a primary cilium (ARL13B$^+$). (D) Whole-mount inguinal white adipose tissue stained for DAPI, ARL13B, EP4, FGFR1OP and the endothelial cell marker CD31 (magenta). Closed arrows highlight EP4 colocalization with primary cilia. (A-C) All data are mean±s.d. and each data point represents an independent experiment. Each independent primary ASC isolation pooled 2-4 mice.

on adipogenesis, and we observed no rescue during PGE2 co-treatments with two versions of a PKA-specific inhibitor, Rp-cAMPs and Rp-8-Br-cAMPs (Fig. 4C). Surprisingly, a third inhibitor of PKA rescued adipogenesis during PGE2 treatment (Fig. 4C). This inhibitor, H89, was also sufficient to rescue

adipogenesis in differentiating isolated primary ASCs co-treated with PGE2 (Fig. S4D).

The discrepancy between PKA inhibitors may be explained by their modes of activity; Rp-cAMPs and Rp-8-Br-cAMPs both inhibit PKA activation by targeting the cAMP binding site on the regulatory

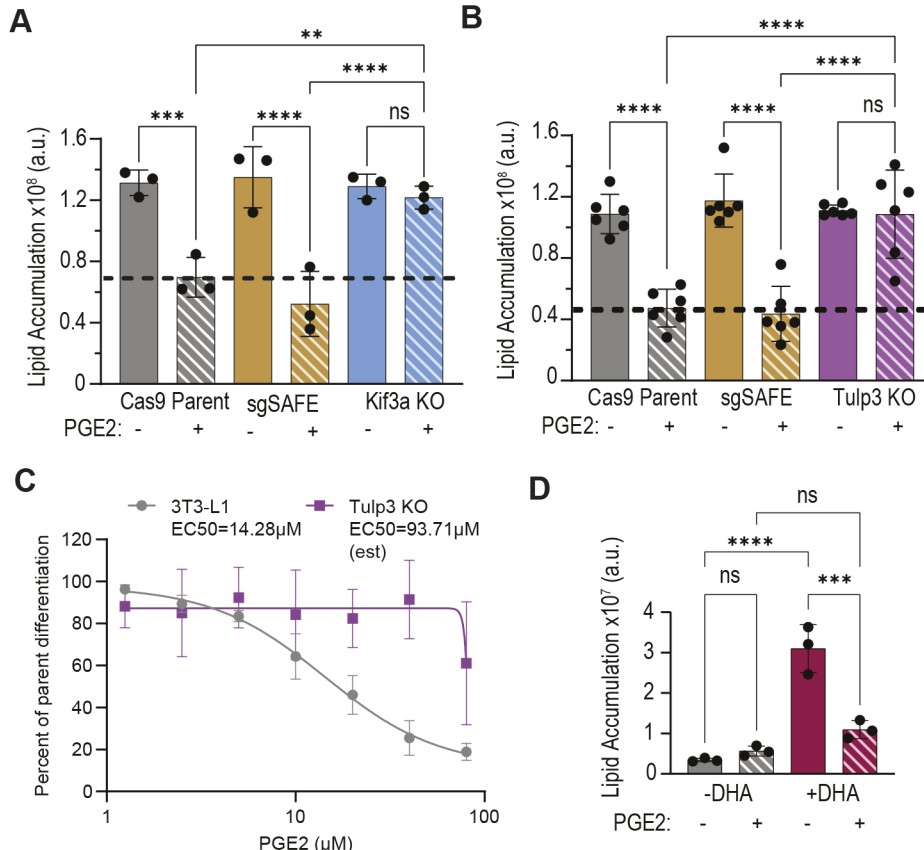

**Fig. 3. Ciliary EP4 is required for PGE2 to inhibit adipogenesis.** (A) Cas9-expressing 3T3-L1 cells, control 3T3-L1 cells expressing a safe sgRNA (sgSAFE) and KIF3A knockout cells differentiated in the presence of 20 µM PGE2 during the first 48 h of adipogenesis. KIF3A knockout cells lacking primary cilia show no response to PGE2. (B) Cas9-expressing 3T3-L1 cells, control 3T3-L1 cells expressing a safe sgRNA, and TULP3 knockout cells differentiated in the presence of 20 µM PGE2 during the first 96 h of adipogenesis. TULP3 knockout cells lacking ciliary GPRCs show no response to PGE2. (C) Dose-dependent response of 3T3-L1 and TULP3 knockout cells to PGE2 during the first 48 h of adipogenesis. Depletion of TULP3 makes 3T3-L1 cells less sensitive to the anti-adipogenic effect of PGE2. Data are mean of three experiments ±s.d. with endpoint lipid content normalized to 3T3-L1 control cell differentiation without PGE2. EC50 calculation for 3T3-L1 and estimation for TULP3 KO performed by log (x) transformation with nonlinear fit regression. (D) 3T3-L1 cells treated with an attenuated differentiation cocktail alone or supplemented with ciliary GPCR agonist DHA and challenged with 20 µM PGE2 during the first 48 h of adipogenesis. PGE2 inhibits adipogenesis in cells treated with the cilia-dependent differentiation cocktail. Endpoint lipid accumulation was recorded after 144 h of differentiation. (A,B,D) Data are mean±s.d.; each data point represents an independent experiment. Dashed lines mark the average endpoint lipid content in control cells treated with PGE2. **$P<0.01$, ***$P<0.001$, ****$P<0.0001$ (one-way ANOVA followed by Tukey's multiple comparison test). a.u., arbitrary units; KO, knockout; ns, not significant.

subunit (Gjertsen et al., 1995), whereas H89 blocks the ATP-binding site on the catalytic subunit of PKA (Engh et al., 1996). All are able to inhibit PKA, although H89 can promiscuously bind to and inhibit several other kinases, albeit to a lesser extent than PKA (Davies et al., 2000). We considered that H89 may rescue PGE2 co-treatment via the off-target inhibition of one or more kinases other than PKA. Using the International Centre for Kinase Profiling, we compiled a list of 17 kinases that retain less than 20% activity in the presence of H89 (Fig. 4D) (Davies et al., 2000). We hypothesized that PGE2 activates one or more of these kinases to inhibit adipogenesis. To identify this target, we tested additional kinase inhibitors targeting discrete subsets of these 17 kinases for the ability to rescue PGE2 activity during adipogenesis (Cabell and Audesirk, 1993; Goodman et al., 2007; Logie et al., 2007; Naqvi et al., 2012; Watanabe et al., 2007). Only two of these inhibitors, Y-27632 and GSK429286A, were able to rescue adipogenesis from PGE2 inhibition (Fig. S4E). Comparing the kinases only inhibited by H89, Y-27632 and GSK429286A, we identified ROCK2 as the most-likely shared target (Fig. S4F). Notably, treatment with the inhibitor GSK429286A more specifically reduced ROCK2 activity

and rescued adipogenesis during PGE2 treatment to a similar degree as H89 (Fig. 4E).

Previous studies have reported that ROCK2 activity can regulate ciliogenesis, cilia length and ciliary signaling (Smith et al., 2025; Streets et al., 2020). We first tested the effect of ROCK inhibitors and PGE2 on 3T3-L1 cilia and observed no differences in the percentage of ciliation or percentage of EP4-positive cilia (Fig. S4G). We then evaluated cilia length under these same treatments. Neither Y-27632 nor GSK429286A treatment alone altered cilia length relative to the control (Fig. S4H). Intriguingly, co-treatment of either inhibitor with PGE2 had no compounding effect on cilia length beyond that of PGE2 alone, despite rescuing PGE2-mediated adipogenic inhibition (Fig. S4H). These data indicate that PGE2 both elongates primary cilia and activates ROCK2, but that changes in length are neither mediated by ROCK2 nor involved in adipogenesis inhibition.

### Ciliary EP4 activates ROCK2 to inhibit adipogenesis
During adipogenesis, the actin cytoskeleton undergoes dramatic rearrangement from stress fiber networks prevalent in ASCs to the

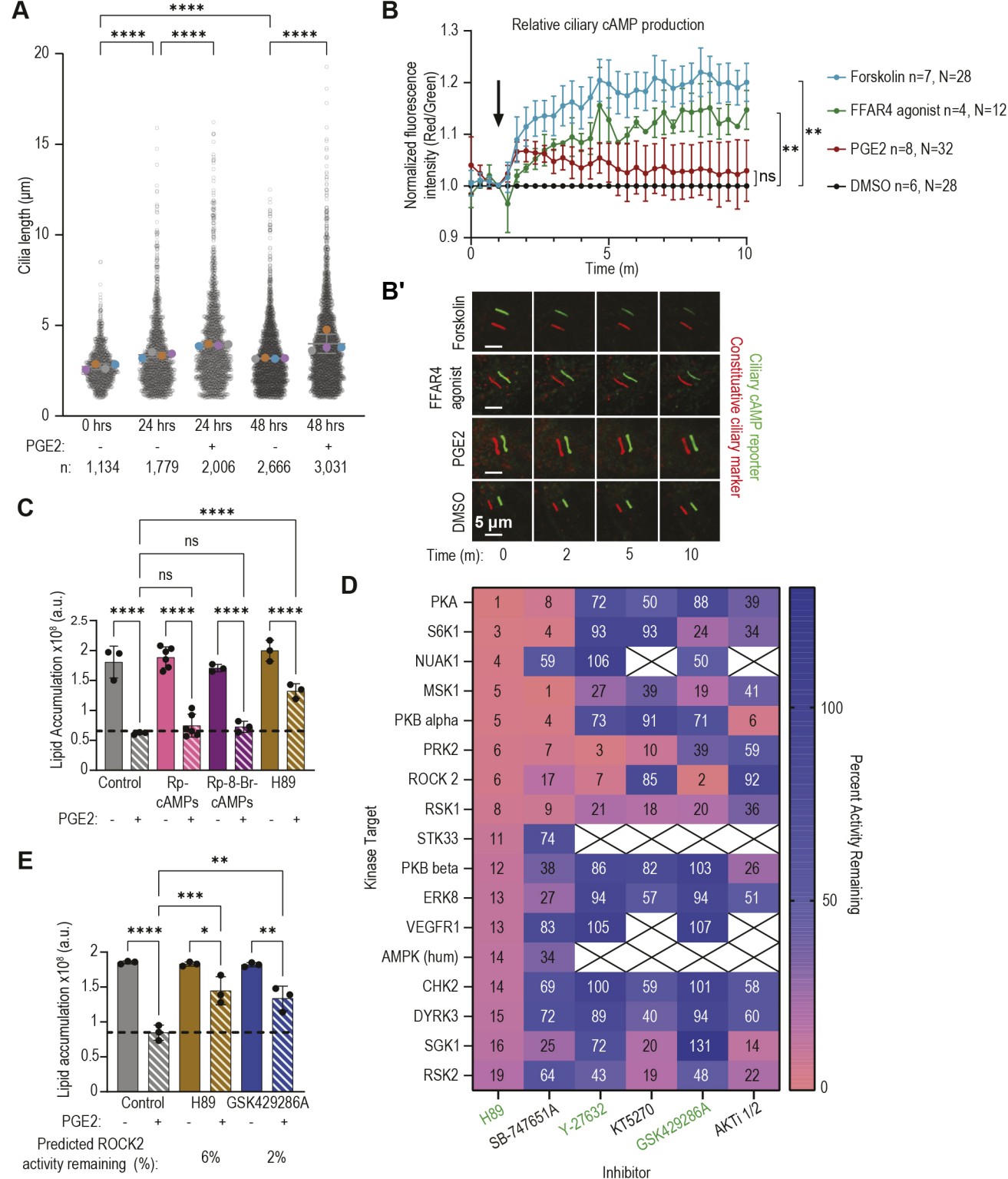

Fig. 4. See next page for legend.

cortical actin network found in mature adipocytes (Mor-Yossef Moldovan et al., 2019). Activation of Rho GTPases, and subsequently ROCK2, stabilizes actin stress fibers which blocks the rearrangement of the cytoskeleton, and this stress fiber retention prevents the progression of adipogenesis and lipid droplet formation (Mor-Yossef Moldovan et al., 2019; Pope et al., 2016). To enable this rearrangement

and consistent with previous reports, we observed that ROCK2 mRNA and protein expression decrease during adipogenesis, although this was not affected by PGE2 treatment (Fig. S5A,B). As ROCK2 is activated by Rho-A (Wei and Shi, 2022), we hypothesized that PGE2 activates ROCK2 via Rho-A to prevent the actin cytoskeleton rearrangement required for adipogenesis.

**Fig. 4. Ciliary EP4 does not signal via ciliary cAMP.** (A) 3T3-L1 primary cilia lengths measured at 0-, 24- and 48-h time points of adipogenesis with or without 20 µM PGE2. Unfilled circles denote lengths of individual cilia measured across four independent trials; filled circles represent the average ciliary length per trial. Mean of all four averages±s.d. shown. *n*=total number of cilia measured per condition across all four trials. (B) Undifferentiated 3T3-L1 cells expressing cilia-targeted cAMP biosensor treated with DMSO (black), the adenylyl cyclase activator forskolin (50 µM, blue), the FFAR4 agonist TUG891 (50 µM, green) or PGE2 (20 µM, red) at the time point indicated by the arrow; images collected every 20 s were used to calculate the ratio of fluorescence intensities between the constitutive ciliary marker and the cAMP sensor at each time point. Data are average of three independent trials ±s.e.m. *n*=total number of experimental wells; *N*=total number of cilia analyzed for each condition. (B′) Representative images of offset cAMP sensor (green) and constitutive ciliary marker (red) at the indicated time points. (C) 3T3-L1 cells treated with different PKA inhibitors (25 µM Rp-cAMPs, 25 µM Rp-8-Br-cAMPs and 10 µM H89) in the presence or absence of PGE2 during the first 96 h of adipogenesis. Dashed line marks the average endpoint lipid content in vehicle-treated control cells treated with PGE2. Only H89 rescues adipogenesis in the presence of PGE2. (D) Heat map of percentage activity remaining for different kinase targets in response to indicated kinase inhibitors. Green text denotes inhibitors that rescue adipogenesis in the presence of PGE2 (see Fig. S4E). (E) 3T3-L1 cells differentiated in the presence or absence of 20 µM PGE2 and the kinase inhibitors H89 (10 µM) or GSK429286A (1 µM) during the first 48 h of adipogenesis. Both inhibitors rescue adipogenesis in the presence of PGE2 and inhibit ROCK2 to similar extents. (C,E) Data are mean±s.d. and each data point represents an independent experiment. *P<0.05, **P<0.01, ***P<0.001, ****P<0.0001 (one-way ANOVA followed by Tukey's multiple comparison test). a.u., arbitrary units; ns, not significant.

We first investigated whether Rho-A transduces the activity of PGE2 in this context. Treatment with a pan-Rho inhibitor coincident with PGE2 partially rescued adipogenesis in 3T3-L1 preadipocytes (Fig. 5A), and treatment with a specific Rho-A activator recapitulated the anti-adipogenic activity of PGE2 (Fig. 5B). To investigate whether PGE2 modulates actin cytoskeleton remodeling, we monitored its effect on actin dynamics during adipogenesis. 3T3-L1 cells were enriched with stress fibers prior to differentiation as measured by phalloidin staining (Fig. 5C), and, consistent with previous reports (Potolitsyna et al., 2024), these stress fibers were completely lost 48 h post-administration of the IDX differentiation cocktail. In stark contrast, actin stress fibers were largely retained when PGE2 was present during adipogenesis (Fig. 5C,D). Co-treatment with the ROCK2 inhibitor GSK429286A rescued this phenotype and resulted in loss of actin stress fibers (Fig. 5C,D). Similarly, Rho inhibition prevented PGE2 from blocking actin rearrangement (Fig. S5C,D). Finally, we confirmed that stress fiber retention during PGE2 treatment is not due to the inability of 3T3-L1 cells to undergo adipogenesis, as treatment with the PPARγ antagonist T0070907 had no effect on actin rearrangement (Fig. S5C,D). Taken together, these data demonstrate that PGE2 inhibits adipogenesis by stabilizing actin stress fibers through the Rho-A/ROCK2 pathway.

Our data show that PGE2 inhibits adipogenesis by signaling through cilia-localized EP4. To determine whether EP4 signaling within primary cilia is necessary to modulate the actin cytoskeleton throughout the cell, we repeated our actin cytoskeleton analysis in TULP3 knockout preadipocytes. As expected, undifferentiated TULP3 knockouts displayed high levels of actin stress fiber networks, and these stress fibers were lost in the first 48 h of adipogenesis (Fig. 5E,F). Unlike in our controls, PGE2 treatment in the TULP3 knockouts did not lead to stress fiber retention, nor did ROCK2, Rho or PPARγ inhibition have any effect on actin cytoskeleton rearrangement in this context (Fig. S5C,D). This is consistent with the inability of PGE2 to

inhibit adipogenesis in TULP3 knockout preadipocytes lacking ciliary EP4. We also observed that induction of adipogenesis with the cilia-dependent DHA cocktail is sufficient to initiate the breakdown of these stress fibers, and that this is prevented by PGE2 treatment (Fig. S5E). Finally, we confirmed that this effect is downstream of EP4 signaling, as the EP4 knockout preadipocytes lost stress fibers regardless of PGE2 treatment during adipogenesis (Fig. 5G,H). We conclude that PGE2 specifically acts upon EP4 localized to the primary cilium of ASCs to activate the Rho-A/ROCK2 signaling cascade and stabilize actin stress fiber networks early in adipogenesis, preventing the necessary cytoskeleton rearrangement required during adipocyte differentiation (Fig. 5I).

## DISCUSSION

White adipose tissue is highly dynamic and expands through both adipocyte hypertrophy and *de novo* adipogenesis to accommodate for increased demands for energy storage in the body. When ASCs fail to undergo adipogenesis, pre-existing adipocytes become hypertrophic and adipose tissue becomes inflamed and fibrotic, leading to adipose tissue dysfunction and increased risk for metabolic disease (Sun et al., 2011). Why ASCs fail to sufficiently undergo adipogenesis in settings linked to metabolic disease, such as obesity, is an active area of investigation. Here, we found that the inflammatory prostaglandin PGE2, previously shown to be elevated in obese adipose tissue (García-Alonso et al., 2016), inhibits ASC early commitment to adipogenesis. PGE2 inhibits ASC differentiation specifically via the GPCR EP4, and we demonstrate for the first time that EP4 localizes to the primary cilium of ASCs *ex vivo* and *in vivo*. The primary cilium and ciliary localization of EP4 are both necessary for PGE2 to inhibit adipogenesis. Activation of ciliary EP4 stimulates the Rho-A/ROCK2 pathway, which stabilizes actin stress fibers and prevents the rearrangement of the cytoskeleton that is necessary in adipogenesis to support rounded adipocyte morphology. Altogether, our data predict that elevated levels of PGE2 in obese and inflamed adipose tissue would further inhibit adipogenesis, perpetuating hypertrophic expansion of adipocytes and driving further adipose tissue inflammation and dysfunction.

Previous studies have investigated the roles of PGE2 and its receptors in adipose tissue physiology (Wang et al., 2022; Xu et al., 2016); however none of these studies interrogated the role of the primary cilium in this context. Primary cilia have emerged as key regulators of cell fate in diverse tissues and cell lineages (Hilgendorf et al., 2024). Almost all stem cells are ciliated and vertebrate development is dependent on proper cilia function (Quinlan et al., 2008). Although cilia constitute less than 1/3000th the volume of a cell, their enrichment with specific receptors and signaling pathway components makes them particularly responsive to extracellular signals (Hilgendorf et al., 2024). Coupled with their proximity to the centrosome, cilia enable rapid, spatially restricted transduction across diverse cellular and physiological contexts (Purkerson et al., 2024). By identifying a previously uncharacterized, cilia-dependent mechanism that restricts cytoskeleton plasticity in ASCs, our data contribute to a growing area of research surrounding the ability of primary cilia to regulate whole-cell morphology and function.

Previous reports demonstrate that ciliary EP4 can activate downstream effectors of the cAMP pathway in retinal epithelial cells, kidney ductal cells and pancreatic beta cells (Ansari et al., 2024; Hansen et al., 2022; Jin et al., 2014; Wu et al., 2021). In contrast, we demonstrate that the stimulation of ciliary EP4 with PGE2 does not lead to the sustained production of ciliary cAMP in ASCs, nor is the anti-adipogenic activity of PGE2 dependent on the activity of downstream effectors of cAMP. The change in ciliary

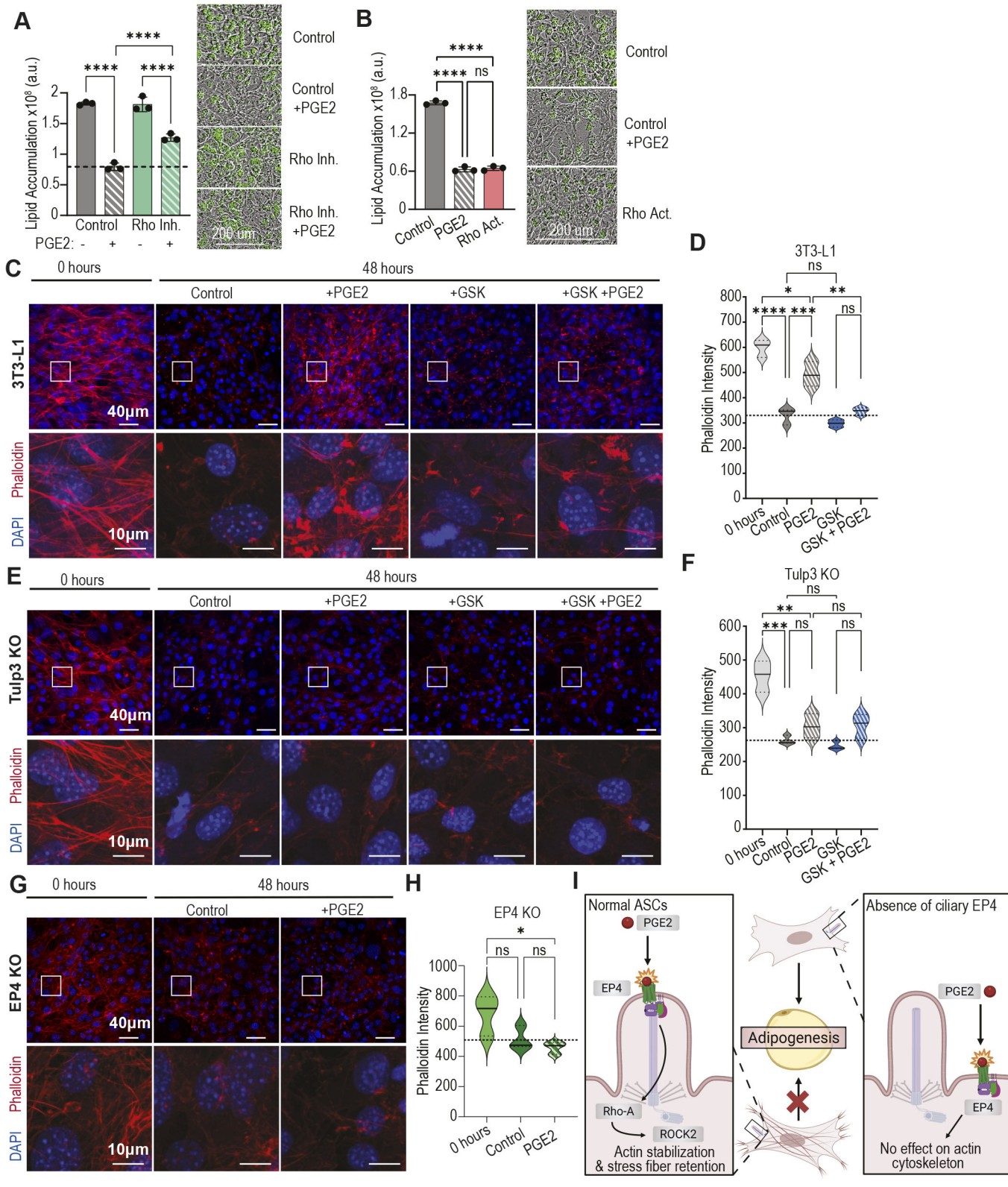

**Fig. 5.** See next page for legend.

cAMP following PGE2 treatment was minimal compared to the activation of the known endogenous ciliary receptor FFAR4, which we previously showed activates adipogenesis via ciliary cAMP production and EPAC activation (Hilgendorf et al., 2019). It seems therefore unlikely that a promotor of adipogenesis, such as FFAR4, and an inhibitor of adipogenesis, such as EP4, would both function by stimulating cAMP levels in the same ciliary compartment. Moreover, a crucial component of the IDX cocktail is the pan-phosphodiesterase inhibitor IBMX; this inhibitor increases global cAMP levels, which is required for the initiation of adipogenesis

**Fig. 5. PGE2 activates ROCK2 in a cilia-dependent manner to inhibit adipogenesis.** (A) 3T3-L1 cells treated with 0.5 µg/ml Rho Inhibitor I during the first 48 h partially rescues PGE2 co-treatment. Left: Endpoint lipid content with dashed line marking average lipid content of vehicle-treated control cells treated with PGE2. Right: Representative images at endpoint of adipogenesis with lipid droplets stained with BODIPY. (B) 3T3-L1 cells treated with 0.25 µg/ml Rho Activator II during the first 48 h of adipogenesis inhibits differentiation with similar efficacy as PGE2. Left: Endpoint lipid content for each condition. Right: Representative images at endpoint of adipogenesis with lipid droplets stained with BODIPY. (C) Actin network of 3T3-L1 cells visualized by phalloidin staining (red). Undifferentiated 3T3-L1 cells predominantly have actin stress fibers, which are disassembled within 48 h of adipogenesis initiation. Stress fibers are not disassembled in the presence of 20 µM PGE2, and stress fiber disassembly is rescued with ROCK2 inhibitor co-treatment (1 µM GSK429286A). Representative images of each treatment condition. Boxed areas are shown at higher magnification below. (D) Violin plot quantifying phalloidin staining intensity in 3T3-L1 cells. (E) Actin networks of TULP3 knockout cells visualized by phalloidin staining. TULP3 knockouts lacking ciliary EP4 disassemble actin stress fibers during adipogenesis regardless of PGE2 treatment or ROCK inhibition. Representative images of each treatment condition. (F) Quantification of phalloidin staining intensity in TULP3 knockout cells. (G) 20 µM PGE2 treatment does not prevent actin stress fiber disassembly in EP4 knockout cells. Representative images of each treatment condition. (H) Quantification of phalloidin staining intensity in EP4 knockout cells. (I) Model for how PGE2 inhibits adipogenesis. In wild-type 3T3-L1 cells and ASCs, activation of ciliary EP4 by PGE2 activates RhoA/ROCK2, which stabilizes the actin cytoskeleton and prevents adipogenesis. Ciliary localization of EP4 is required for this activity. Created in BioRender by Lee, M., 2025. https://BioRender.com/t962fv5. This figure was sublicensed under CC-BY 4.0 terms. (C,E,G) Boxed areas are shown at higher magnification below. (D,F,H) Violin plots quantify phalloidin staining intensity from 18 images per condition collected equally across three independent repeats. Solid lines within each violin depict the median, while dotted lines are quartiles. Dashed lines across the graph indicate the average phalloidin staining in vehicle treated control cells after 48 h of adipogenesis. (A,B) Data are mean±s.d. and each data point represents an independent experiment. (A,B,D,F,H) *$P<0.05$, **$P<0.01$, ***$P<0.001$, ****$P<0.0001$ (one-way ANOVA followed by Tukey's multiple comparison test). a.u., arbitrary units; ns, not significant.

through EPAC activation (Ji et al., 2010; Petersen et al., 2008). While we cannot entirely exclude the possibility that the transient increase in cAMP following PGE2 treatment is mediating its inhibition of adipogenesis, it seems more likely that ciliary EP4 functions through alternative, non-cAMP transducers to activate Rho-A/ROCK2 signaling in this context.

Previous studies have shown that the EP4 receptor can both activate and inhibit the Rho/ROCK2 pathway (Khan and Ghosh, 2025; van Helden et al., 2008; Zhang and Daaka, 2011), implying context-specific regulation of ROCK2 activity by PGE2. While ROCK2 has been shown to regulate ciliogenesis and cilia length (Smith et al., 2025; Streets et al., 2020), we show here that ROCK2 inhibition is insufficient to regulate either in 3T3-L1 preadipocytes (Fig. S4G,K), although it remains unclear whether ROCK2 activation has an effect on ciliary composition beyond EP4. As cilia length increases are most often associated with increased ciliary cAMP and PGE2 treatment shows a short-lived increase in ciliary cAMP, EP4 may couple to multiple heterotrimeric G protein complexes, including but not limited to $G_{\alpha s}$, to lengthen cilia in ASCs. However, the downstream activation of canonical cAMP effectors is not involved in the anti-adipogenic activity of PGE2. Thus, PGE2 may activate multiple downstream mechanisms that independently drive cilia length changes and inhibit adipogenesis through ROCK2 activation and stress fiber stabilization.

Inhibition of the Rho-A/ROCK2 pathway is essential during adipogenesis as it allows for the cytoskeletal restructuring in differentiating adipocytes that is required to accommodate their

growing lipid droplets (Pope et al., 2016). ROCK2 activity is mediated by interactions with Rho-GTPases, typically Rho-A (Hartmann et al., 2015). Canonically, the Rho-A, -B and -C GTPases are activated by specific guanosine nucleotide exchange factors (GEFs), of which 69 unique Rho-GEFs have been identified in humans (Rossman et al., 2005). These Rho-GEFs are activated by GPCRs coupled to $G_{\alpha 12/13}$ transducers, although alternative activation pathways, such as those involving the $G_{\alpha q}$ family, have also been reported (Mao et al., 1998). In other cases, Rho-GEFs may be activated by the $G_{\beta \gamma}$ complex (Niu et al., 2003), highlighting the diversity of signal transduction within a single pathway. We observed that ciliary EP4 stimulation ultimately activates ROCK2 in ASCs, although through which G protein or Rho-GEF is not yet known. Studies have highlighted that EP4 can promiscuously couple to several $G_{\alpha}$ subfamilies (Masuho et al., 2023), potentially allowing for Rho-A activation by $G_{\alpha 12/13}$ or $G_{\alpha q}$ transduction; alternatively, this mechanism may be mediated entirely by the $G_{\beta \gamma}$ subunit or an alternative signaling pathway. Finally, oligomerization of GPCRs, as has been reported for EP receptors (Barnes, 2006; Ferré et al., 2014), may enable uncharacterized downstream signaling cascades.

Several noteworthy features of the ciliary microenvironment could enable GPCR oligomerization and other types of crosstalk, including the high density of GPCRs and intracellular effectors within close proximity inside the cilioplasm. Crosstalk between ciliary receptors has been documented, such as the reduction in cAMP response from the D1 dopamine receptor following the activation of another ciliary receptor, GPR88 (Marley et al., 2013), and the attenuation of melanin concentrating hormone's effect in neurons treated with a Smoothened agonist (Bansal et al., 2019). Broadly speaking, the compartmentalization of signals in the constrained ciliary volume with only a single diffusible outlet may limit the transduction capacity of individual pathways in isolation. As such, cilia may function much like a molecular rheostat, consolidating multiple extracellular signals into a single output gated by the centrosome in the cell body. The sum of all receptors within a primary cilium would be determinant in their ability to convey extracellular messages, and heterogeneity between the 'ciliary fingerprint' of cells within or between tissues would drastically alter their responses to coeval signals. As such, any disruption of ciliary composition would be detrimental to the adipogenic potential of ASCs, as seen in the TULP3 knockout preadipocytes that fail to undergo adipogenesis without the supraphysiological differentiation cocktail (Hilgendorf et al., 2019). Ultimately, future investigation is required to uncover the molecular basis of how ciliary EP4 activation in ASC cilia drives Rho-A/ROCK2 activity.

The localization of EP4 to ASC primary cilia opens a potential avenue for therapeutic intervention to combat hypertrophic adipose tissue expansion. Current pharmaceutical interventions, such as incretin mimetics, are incredibly efficient for weight loss and mitigating metabolic disease, but they are only effective during their treatment periods and do not treat the underlying mechanisms of metabolic dysfunction (Phuong-Nguyen et al., 2024; Wilding et al., 2022). Thus, there is immense value in investigating novel and/or auxiliary therapeutics that target physiological regulators of adipose tissue health and expansion. As our work highlights, one approach may be to prevent excessive PGE2 production in order to promote *de novo* adipogenesis. Nonsteroidal anti-inflammatory drugs (NSAIDs) potently inhibit prostaglandin synthesis and are one of the most commonly used drug classes internationally (Brennan et al., 2021). The use of NSAIDs has been associated with decreased

risk for type 2 diabetes (Lin et al., 2022; Thomas et al., 2003), and a systematic review of the effects of NSAIDs on white adipose tissue found that their administration overwhelmingly reduced adipocyte hypertrophy and chronic inflammation (da Cruz Nascimento et al., 2022). Other studies even showed that NSAIDs can directly increase adipogenesis in vitro (Cacciatore et al., 2023; Lehmann et al., 1997; Puhl et al., 2015). These anti-diabetic effects of NSAIDs are likely to be a product of numerable direct and indirect signaling changes, both within adipose tissue and throughout the body, and this multi-factorial, coordinated activity is an appealing factor in the usage to combat metabolic disease. While it is difficult to identify the relative contribution of ciliary EP4 signaling in ASCs to the overall anti-diabetic effect of NSAIDs, our study has highlighted a mechanism by which lowering the prostaglandin PGE2 may help promote metabolic health. Moreover, the targeting of ciliary GPCRs, such as EP4 and other receptors yet to be elucidated, might present a valuable approach to preserve adipose tissue function and metabolic health in individuals with obesity by augmenting the rate of adipogenesis and promoting healthier adipose tissue expansion.

## MATERIALS AND METHODS
### 3T3-L1 adipogenesis and treatments
3T3-L1 cells acquired from ATCC were grown and maintained in DMEM supplemented with 10% bovine calf serum, 1% penicillin/streptomycin and 1% GlutaMax (DMEM-BCS). For adipogenesis, cells were plated on 96-well plates at $4.5e^3$ cells per well and grown to confluency arrest for 96 h, with a media change after 48 h. To initiate differentiation, cells were treated with the IDX differentiation cocktail in DMEM containing 10% fetal bovine serum, 1% penicillin/streptomycin and 1% GlutaMax (DMEM-FBS), for a final concentration of 2 µg/ml insulin, 1 µM dexamethasone and 0.5 mM 3-isobutyl-1-methylxanthine (IBMX). Cells were also supplemented with 0.5 µg/ml BODIPY 493/503 to measure intracellular lipid accumulation. The IDX media remained on cells for the first 48 h of adipogenesis, after which it was replaced with a freshly made adipocyte maintenance media containing 1 µg/ml of insulin and 0.5 µg/ml BODIPY 493/503 in DMEM-FBS. Additional maintenance media changes occurred every 48 h thereafter. The attenuated differentiation cocktail used to assess cilia-dependent adipogenesis was composed of 0.4 µg/ml insulin, 0.1 µM dexamethasone and 0.02 mM IBMX supplemented with 100 µM DHA.

Lipid accumulation was monitored using the Satorius IncuCyte Live-Cell Analysis System (Essen BioScience), capturing phase and channel 488 fluorescent images at 10× magnification every 4-8 h. Lipid accumulation was measured by measuring the total integrated fluorescence intensity at 488 nm. All adipogenesis endpoint measurements were collected after 6 days of differentiation unless otherwise stated. For treatments, 20 µM of PGE2 resuspended in DMSO or a DMSO vehicle was added to the IDX media and the first maintenance media treatment unless otherwise stated. Antagonists of the PGE2 receptors EP1-4 were added 24 h prior to IDX and PGE2 treatments and again during the first 48 h of IDX stimulated differentiation. All inhibitors and activators of PKA, EPAC, assorted kinases, and Rho (Table S1) were present during the first 48-96 h of adipogenesis, and co-treated with PGE2 as indicated. All cell lines used were tested for contamination.

### Primary cell isolation and adipogenesis
For each independent experiment, the stromal vascular fraction was isolated from white adipose tissue from two to four C57BL/6J mice as previously reported (Rodeheffer et al., 2008). All mice were between 10 and 16 weeks. This fraction was incubated with antibodies specific to CD45 (Invitrogen, 25-0451-82), CD31 (Invitrogen, 25-0311-82), Ter119 (Invitrogen, 25-5921-82), CD34 (BD Pharmingen, 551387), Sca1 (BioLegend, 108112) and CD29 (Invitrogen, 11-0291-82) for 30 min at a 1:150 dilution. Adipocyte stem cells were isolated using FACS for lineage negative (CD45− CD31− TER119−) and positive (CD34+ SCA1+ CD29+) markers. Cells were plated at $1.5e^4$ cells per well in 96-well plates in

DMEM-FBS and until confluency was reached, approximately 96-120 h after plating. Confluent ASCs were then treated with the IDX differentiation cocktail with BODIPY 493/503 for 72 h, then maintained in adipocyte maintenance media every 48 h thereafter. Lipid accumulation was monitored via the IncuCyte system. At the same time as the IDX cocktail, 20 µM of PGE2 suspended in DMSO or a DMSO vehicle was added to the cells. Any co-treatment was added coincident with the IDX cocktail and PGE2. All use of murine models were approved under the University of Utah's Institutional Animal Care and Use Committee.

### Quantitative PCR
RNA was isolated from control undifferentiated 3T3-L1 cells or cells treated with the IDX cocktail for 96 h in the presence of 20 µM PGE2 or equivalent DMSO vehicle lysed with QIAzol Lysis Reagent according to the RNeasy Lipid Tissue Mini Kit (QIAGEN, 74804). One to two micrograms of RNA was used to generate cDNA using M-MLV RT (Invitrogen, 28025-013) and 5X FS buffer (Invitrogen, Y02321) according to the manufacturer's instructions. TaqMan probes targeting Pparg (Thermo Fisher Scientific, MM01184322_M1), Cebpa (Thermo Fisher Scientific, Mm00514283_s1), Adipoq (Thermo Fisher Scientific, Mm04933656_m1), Fabp4 (Thermo Fisher Scientific, Mm00445878_m1), Rock1 (Thermo Fisher Scientific, Mm00485733_m1), Rock2 (Thermo Fisher Scientific, Mm01270843_m1) and the housekeeping gene Nono (Thermo Fisher Scientific, MM00834875_G1) were amplified using TaqMan Gene expression master mix (Thermo Fisher Scientific, 4369016) in MicroAmp Optical 384-well reaction plates (Applied Biosystems, 4309849) according to manufacturer protocols. Expression was analyzed on the QuantStudio™ 7 Flex Real-Time PCR System (Thermo Fisher Scientific, 4485701). All samples were run in biological and technical triplicate; CT values were averaged across technical replicates and normalized to Nono expression for each time point, and individual probes were normalized to their respective expression on day 0.

### Crispr/Cas9 small guide RNA-mediated mutant generation
3T3-L1 Cas9-expressing cell lines were generated by infection with plentiCas9-Blast (Addgene, plasmid #52962) and pMCB320 (Addgene plasmid #89359) plasmids (deposited by M. Bassik, Stanford University). Small guide RNAs targeting the Ptger4 or Kif3a locus were ordered through Integrated DNA Technologies as short DNA oligos (Table S2) and annealed as dsDNA oligos for ligation into target the in-frame GFP lentiviral target vector pMCB320 following digestion with BstXI and BlpI. HEK293T cells were transfected with the guide-containing vector, pCMV delta R8.2 dvpr (Addgene plasmid #8455), and pCMV VSV-G (Addgene plasmid #8454) using OptiMEM and FuGENE 6 Transfection Reagent (Promega, E2691) according to manufacturer protocols. Viral supernatant was harvested after 48 h and used to infect 3T3-L1 cells stably expressing Cas9-BFP in antibiotic-free DMEM-BCS. Infected cells were then sorted for BFP+GFP+ cells. Genomic extract was used in PCR to amplify the targeted cut site for each knockout and sent for Sanger sequencing (Table S2). Sequencing was then analyzed by TIDE sequence analysis (Brinkman et al., 2014) (https://tide.nki.nl/) to determine knockout efficiency.

### Immunoblotting
Confluent 3T3-L1 cells were harvested from 6-well plates using 100 µl NuPAGE 4× LDS Sample Buffer loading dye (Thermo Fisher Scientific, NP0008) per well. Cell lysate was transferred to QIAshredder tubes (QIAGEN, 79656) and processed per manufacturer protocols. Equivalent samples were loaded into NuPAGE 4-12% Bis-tris Gels (Thermo Fisher Scientific, NP0322BOX) and run in 1× MES Running Buffer (Thermo Fisher Scientific, NP0002) at 200 V. Proteins were transferred to nitrocellulose membranes (Thermo Fisher Scientific, 88018) for 1 h at 4°C in transfer buffer [25 mM Tris, 192 mM glycine, 20% (v/v) methanol, pH 8.3 in Milli-Q water] at 20 V. Membranes were then blocked in 5% skim milk in TBS with 0.1% Tween 20 (TBST) for 1 h, washed three times with TBST for 10 min each, and immunostained with primary antibodies in 1% skim milk in TBST with shaking overnight at 4°C. Blots were washed three more times as before, stained with secondary LI-COR antibodies at 1:20,000

Journal of Cell Science

in 1% skim milk and 0.1% SDS TBST for 1 h at room temperature. Blots were washed three times as previous prior to imaging on the LI-COR Odyssey CLX. Antibodies targeting α-tubulin (Santa Cruz Biotechnology, sc-32293; 1:1000) and β-actin (Cell Signaling Technology, 4970; 1:1000) were used as loading controls, while antibodies targeting EP1 (Thermo Fisher Scientific, BS-6316R; 1:1000), EP2 (Abcam, AB167171; 1:1000), EP3 (Thermo Fisher Scientific, 14357-1-AP; 1,1000), EP4 (Santa Cruz Biotechnology, sc-55596; 1:200), KIF3A (Proteintech, 13930-1-AP; 1:1000) and ROCK2 (Cell Signaling Technology, 8236S; 1:1000) were used to identify proteins of interest. For ROCK2 expression, 100,000 3T3-L1 cells were plated in a 6-well dish to reach confluency for differentiation. Cells were treated at day 0 with the differentiation cocktail with or without 20 µM PGE2. At 0 h, 48 h and 96 h, cells were collected in 4× LDS Sample Buffer and processed as above. Quantification of protein expression was carried out in the LI-COR Odyssey CLX program by normalizing target protein intensity to its respective loading control, then normalizing the expression in the knockout to the expression in wild-type 3T3-L1 cells. All uncropped blots are available in Fig. S6.

### Immunofluorescence imaging

3T3-L1 cells or isolated ASCs from white adipose tissue were plated on acid washed 12 mm coverslips (Electron Microscopy Sciences, 72230-01) or 16-well glass chamber slides (Lab-Tek, 178599). Confluent cells were fixed for 10 min with 4% paraformaldehyde, then blocked in 5% goat and 5% donkey serum resuspended in immunofluoresence (IF) buffer (PBS with 3% bovine serum albumin, 0.1% NP-40 substitute and 0.02% sodium azide). Cells were then stained for 1-4 h with antibodies targeting EP4 (Santa Cruz Biotechnology, sc-55596; 1:200), FGFR10P (Proteintech, 11343-1-AP; 1:500) and ARL13B (NeuroMabs, 75-287; 1:1000). In special cases, cells were stained with antibodies specific to PCM1 (Santa Cruz Biotechnology, sc-398365; 1:1000) or acetylated tubulin (Proteintech, 66200-1-Ig; 1:200). Secondary antibodies [Thermo Fisher Scientific, Alexa Fluor Plus Highly Cross-Adsorbed Secondary Antibodies (various)] were applied at 1:2000 dilutions in IF buffer for 30 min, and DAPI dye was applied at 2 µg/ml for 5 min at room temperature. Stained cells were imaged using either a Delta Vision Ultra Widefield or Nikon Ring Spinning Disk confocal microscope at 40-60× magnification. Images were analyzed in ImageJ and manually counted for cilia frequency and EP4 localization to cilia.

### Whole-mount imaging

Live C57BL/6J mice were anesthetized continuously with isoflurane within a fixed chamber. The abdominal cavity was opened and the right atrium of the heart punctured. A syringe was inserted into the left ventricle and used to steadily profuse 5-10 ml of PBS through the tissue, immediately followed by 5 ml of 4% paraformaldehyde. Inguinal and perigonadal white adipose tissue was collected and gently cut into 1-2 mm strips then transferred into 4% paraformaldehyde followed by 0.3% Triton-X in PBS (PBST) for 15 min at room temperature. Samples were then washed three times for 20 min each in 0.3% PBST. Samples were incubated treated with 20 µg/ml of Proteinase K in 10 mM Tris-HCl buffer (pH 7.4) for 5 min before being transferred to chilled 100% methanol for 30 min. Samples were washed three times for 20 min each in PBST and moved into blocking solution containing 3% normal donkey serum and 3% normal goat serum in PBST overnight at 4°C. Samples were incubated with primary antibody suspended in PBST for 6 h at room temperature, washed three times with PBST for 30 min each, then incubated with secondary antibodies at 1:500 in PBST overnight at 4°C. Samples were then incubated with 2 µg/ml Hoechst 33342 dye (Thermo Fisher Scientific, 62249) and, if indicated, HCS LipidTOX (Thermo Fisher Scientific, H34477; 1:200) in PBST for 30 min. Samples were washed twice more with PBST for 30 min each, cut into ~2 mm³ chunks, and mounted onto glass slides with Glycergel Mounting Media (Dako, C056330-2) containing 0.02 mg/ml 1,4-diazabicyclo[2.2.2]. Stained tissue was imaged using the Nikon Ring Spinning Disk confocal microscope at 60× magnification. Images were captured at 0.3 µm z-step increments for 18-54 steps, then analyzed by 3D image rendering or maximum intensity projections using ImageJ. Antibodies used were as follows: anti-CD31 (BD Biosciences, 553370; 1:200), anti-ARL13B (NeuroMabs, 75-287; 1:1000),

anti-FGFROP (Proteintech, 11343-1-AP; 1:500), anti-EP4 (Santa Cruz Biotechnology, sc-55596; 1:200).

### Cilia length measurements

3T3-L1 cells were fixed prior to, or 24 or 48 h after, treatment with the IDX cocktail and 20 µM PGE2 or DMSO vehicle. For ROCK2 inhibition experiments, cells were fixed 48 h after treatment with the IDX cocktail containing Y-27632 (20 µM) or GSK429286A (1 µM), with and without PGE2 (20 µM). Cells were stained as indicated above with antibodies targeting ARL13B and FGFR1OP, as well as with Hoechst 33342 dye. Images were captured at 60× using a Nikon Ring Spinning Disk confocal microscope, and imported into ImageJ. Images were analyzed using the ImageJ plug-in CiliaQ (Hansen et al., 2021) with Canny 3D segmentation to measure cilia length through the x-, y- and z-planes for each identified cilium. Cilia that were in contact with any planal bounds were excluded from the analysis, and a manual minimum length requirement of 1 µm was applied as a cut-off. Length measurements were collected over four independent trials.

### Cilia-targeted cAMP biosensor imaging

3T3-L1 cells were plated in 16-well CultureWell coverglass slides (Grace BioLabs, 112353) at 1e⁴ cells per well in DMEM-BCS then grown for 24 h at 37°C. The following day, cells were infected with the Ratiometric Cilia-targeted cADDis cAMP green-down sensor (Montana Molecular, D0211G) baculovirus at a 1:6 dilution in DMEM-BCS supplemented with 2 mM sodium butyrate for 30 min at room temperature protected from light. Next, they were incubated at 37°C for 6 h before replacing their media with DMEM-BCS containing 2 mM sodium butyrate. After an additional 24 h of incubation at 37°C, the cells were used for live cell imaging; 10 min prior to imaging, cell media was replaced with 100 µl warmed PBS. For each well, three to nine cilia were identified and marked for imaging. Each cilium was imaged for both the cAMP reactive fluorophore (488 nm) and the constitutive fluorophore (561 nm) every 20 s using the Nikon Ring Spinning Disk confocal microscope. Cilia were imaged for 1 min without treatment to measure baseline fluorescence intensities and bleaching. After 1 min, PBS containing individual treatment conditions were added to the wells to bring the final concentrations to 20 µM PGE2, 50 µM forskolin or 50 µM TUG891. All compounds were suspended in DMSO, and control wells received PBS with DMSO only. Cilia that drifted out of focus during the time course were excluded from the analysis. The ratio of red (constitutive) to green (cAMP-reactive) fluorescence was recorded for each cilium. To normalize basal differences in ciliary fluorescence, fluorescence intensity for each cilium was normalized to the last reading preceding treatment. Cilia from each technical repeat well were averaged into a single time course per well, and each replicate well was averaged to a single data point for each independent biological repeat. Biological repeat averages were normalized to their respective DMSO controls to account for photobleaching. Data are presented as the average of three independent repeats, with n representing the total number of wells for each treatment and N indicating the total cilia for recorded from each condition.

### Actin staining and intensity recording

3T3-L1 cells were plated at 4.5e³ cells per well on 16-well chamber slides and grown for 96 h to confluency arrest. Control untreated cells were fixed with 4% paraformaldehyde while remaining chamber slides were treated with the IDX cocktail and the indicated co-treatments for 48 h. Treated cells were fixed as before, and all chamber slides were stained with the actin cytoskeleton dye phalloidin 565 nm (Thermo Fisher Scientific, A12380) and DAPI as per the manufacturer's recommendations. Cells treated with the cilia-dependent differentiation cocktail containing DHA were fixed after 36 h. Cells were imaged using the Nikon Ring Spinning Disk confocal microscope at 60× magnification. ImageJ was used to generate z-stack maximum intensity projections and the total fluorescence intensity of phalloidin staining was measured with ImageJ mean intensity measurements.

### Acknowledgements
Research reported in this publication utilized the Cancer Bioinformatics Shared Resource at Huntsman Cancer Institute at the University of Utah and was supported

by the National Cancer Institute of the National Institutes of Health under Award Number P30CA042014. This core facility provided support and guidance in the analysis of the single nucleus RNA transcriptomics data reported in Emont et al., 2022. The content is solely the responsibility of the authors and does not necessarily represent the official views of the NIH. We acknowledge the Cell Imaging Core at the University of Utah for use of equipment and the Flow Cytometry Core at the University of Utah for use of equipment. Schematics were created in BioRender.com. We thank Dr Yang Liu for assistance for all cell imaging with the Nikon spinning disk confocal microscope.

## Competing interests
The authors declare no competing or financial interests.

## Author contributions
Conceptualization: K.I.H.; Formal analysis: M.D.L.; Funding acquisition: M.D.L., K.I.H.; Investigation: M.D.L.; Methodology: M.D.L.; Resources: K.I.H.; Visualization: M.D.L.; Writing – original draft: M.D.L.; Writing – review & editing: K.I.H.

## Funding
This work was supported by the National institute of Diabetes and Digestive and Kidney Diseases (R01DK133455 to K.I.H.; Ruth L. Kirschstein National Research Service Awards 5T32DK091317 and F31DK139738 to M.D.L.). Additional support was through the Pew Charitable Trusts (K.I.H.). Open Access funding provided by the University of Utah. Deposited in PMC for immediate release.

## Data and resource availability
All relevant data and details of resources can be found within the article and its supplementary information.

## First Person
This article has an associated First Person interview with the first author of the paper.

## Peer review history
The peer review history is available online at https://journals.biologists.com/jcs/lookup/doi/10.1242/jcs.264193.reviewer-comments.pdf

## Special Issue
This article is part of the Special Issue 'Cilia and Flagella: from Basic Biology to Disease', guest edited by Pleasantine Mill and Lotte Pedersen. See related articles at https://journals.biologists.com/jcs/issue/138/20.

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
