## [Peer Review File · Journal of Cell Science]

Prostaglandin E2 inhibits adipogenesis through the cilia-dependent activation of ROCK2

Mark D. Lee and Keren I. Hilgendorf
DOI: 10.1242/jcs.264193

Editor: Lotte Pedersen

Review timeline

Original submission:	6 June 2025
Editorial decision:	1 July 2025
First revision received:	24 August 2025
Accepted:	8 September 2025

Original submission

First decision letter

MS ID#: jcs.264193

MS TITLE: Prostaglandin E2 inhibits adipogenesis through the cilia-dependent activation of ROCK2
AUTHORS: Mark D. Lee; Keren I Hilgendorf

ARTICLE TYPE: Research Article

Dear Dr Hilgendorf,

We have now reached a decision on the above manuscript.

To see the reviewers' reports and a copy of this decision letter, please go to:

As you will see, the reviewers raise a number of substantial criticisms that prevent me from accepting the paper at this stage. They suggest, however, that a revised version might prove acceptable, if you can address their concerns. If you think that you can deal satisfactorily with the criticisms on revision, I would be pleased to see a revised manuscript. We would then return it to the reviewers.

Reviewer 1

Advance summary and potential significance to field

In this study, Lee & Hilgendorf showed that PGE2 inhibit adipogenesis of adipocyte stem cells (ASCs) and 3T3-L1 cells induced by insulin+dexamethasone+IBMX (IDX cocktail). Using specific inhibitors and KO cells, the authors showed that this inhibition is mediated by the EP4/PTGER4 receptor. In addition, using Kif3a-KO cells and Tulp3-KO cells, the EP4-mediated inhibition was shown dependent on cilia and suggested to be dependent on ciliary localization of EP4. Furthermore, the authors showed that the EP4-mediated inhibition is not dependent on canonical cAMP signaling, but, unexpectedly/curiously, dependent on activation of the Rho-ROCK2 pathway, resulting in stabilization of the stress fiber. Although the mechanism linking EP4 and the Rho-ROCK2 pathway is unclear, I think this paper is worthy of publication in JCS as the findings in this study are novel. However, it is a bit puzzling in

light of previous studies involving one of the authors (Hilgendorf KI), and the concerns described below need to be addressed.

Comments for the author

Major points:

1. Agonists of FFAR4/GPR120 and those of EP4 were shown to stimulate secretion of insulin and glucagon via cAMP signaling (Wu et al., 2021). In addition, Hilgendorf et al. (2019) showed that omega3-FA stimulates ciliary FFAR4 to induce adipogenesis in 3T3-L1 cells via the cAMP pathway. Although the present study showed that stimulation of 3T3-L1 cells with FFAR4 agonist but not PGE2 induces cAMP production, it was not investigated what happens to adipogenesis; namely, the question is whether PGE2 inhibits adipogenesis in 3T3-L1 cells induced not only by IDX but also by FFAR4 agonists. In addition, can FFAR4 agonists inhibit stress fiber formation? If so, can PGE2 cancel the FFAR4-mediated inhibition of stress fiber formation? If PGE2 inhibits FFAR4-stimulated adipogenesis, discuss what discriminates between Gs-coupling of FFAR4 and Gs-uncoupling of EP4 in 3T3-L1 cells.
2. The authors showed that ROCK inhibitors (Y-27632 and GSK429286A) antagonize EP4-mediated inhibition of adipogenesis and remodeling of stress fibers. However, the underlying mechanism is unclear. To confirm that these inhibitors do not affect the formation of cilia or the entry of EP4 into cilia, data are needed to verify the rate of cilia formation (cilia length) and the ciliary localization of EP4 upon the inhibitor treatment.

Minor points:

1. Page 7, line 52: Tulp3 is not a motor protein, but is an adaptor protein linking the IFT-A complex and GPCRs (for example, see Mukhopadhyay, et al., Genes Dev., 2010; Badgandi, et al., JCB, 2017; Hirano, et al., MBoC, 2017).
2. Most recently, it was shown that ROCK2 may be involved in ciliogenesis by affecting the transport of ciliated vesicles via actin cytoskeleton remodeling (Smith, et al., Commun. Med. 2025). Add this to Discussion.

Reviewer 2

Advance summary and potential significance to field

In this manuscript by Lee and Hilgendorf, the authors discover another cell type - the adipocyte stem cells, in which the E-type prostaglandin receptor 4 (EP4) localizes to primary cilia. Within their manuscript, the previously known function of PGE2 and EP4 in adipogenesis is directed by the authors into the ciliary compartment. Herein, the authors are expanding on cell fate mechanisms that are regulated by the primary cilia for fine tuning, and offering insights on how chronic inflammation may result in adipose tissue dysfunction and metabolic disease progression. While the manuscript introduces novelty, I have several comments that need to be addressed in order to further support the statements authors made.

Comments for the author

1. P6, l13-17: "PGE2 completely abrogates the differentiation of ASCs isolated from the inguinal white adipose depot of both male and female lean mice (Fig. 1 E, F, FigS1F, G) and similarly inhibits adipogenesis in perigonadal ASCs (Fig S1G-I)" This is not true for the female perigonadal ASCs (Fig. S1I). This appears to correlate with the rather small difference in EP expression of obese vs. lean mice and humans (Fig. S1A, B). The authors should discuss this sex-dependent difference.
2. A gene knock-out is introduced at several instances in the manuscript (for EP4, KIF3A and TULP4). The authors need to show protein levels of the targeted proteins, such as by using western blot, to make sure about the loss of protein or possible truncations. A related question - if a complete EP4 knock-out was induced, how do you explain EP4+ cilia in these cells (Fig. S2A, B)?
3. P8, l20-24: "Consistent with previous findings (Dalbay et al, 2015), cilia length transiently increases 24 hours post-initiation of adipogenesis with the IDX cocktail (Fig. 4A, Fig S4A). Addition

of PGE2 results in a greater and more persistent increase in cilia length...". The authors do not show any timely dynamics of the cilia length; in the two time points they use, there is no difference in cilia length. Hence, they cannot talk about "transient increase" or "more persistent increase" in cilia length, since they only see elongation that does not go back.

4. Fig. 3B: Knock-out of TULP3 produces partial inhibition of ciliary EP4 localization, resulting in complete reversal of the inhibitory effect of PGE2 on adipogenesis. This raises a possibility that other receptors and signaling pathways may have been affected, contributing to the loss of adipogenesis phenotype. This needs to be thoroughly discussed in the manuscript.

5. The conclusions on cAMP signaling having negligible-to-no role in the PGE2/EP4-mediated adipogenesis block seem preliminary. Fig. 4B shows that a single tested concentration of PGE2 produces lesser effect on ciliary cAMP production than a single tested concentration of FFAR4 agonist and forskolin. The effects can, however, be concentration-dependent. The authors need to test at least three concentrations of the respective drugs, in order to show that the PGE2 concentration used throughout the experiment to inhibit adipogenesis indeed produces a relatively minor ciliary cAMP, compared to FFAR4 agonist and forskolin.

6. Fig. 5A: mRNA levels were quantified in cells treated or not with PGE2 during differentiation. In order to make sure that protein levels also drop in a similar way, the authors need to run ROCK1/2 western blot.

7. Fig. 5A: The representative images need higher magnification to fully appreciate the phenotype. With the current images, I see no difference between Rho Inh. and Rho Inh.+PGE2 samples.

8. Finally, to further support their model (Fig. 5F), the authors need to provide additional experiments targeting the ciliary PEG2/EP4-Rho-A-ROCK2-actin cytoskeleton axis. How is cytoskeleton remodeling (analogous to experiments in Fig. 5C, D) in the EP4-knock-out 3T3-L1 cells? Do you achieve similar results with genetic/epigenetic (CRISPR/siRNA) targeting of ROCK2 (to eliminate the controversy of using small molecule inhibitors)?

9. A general comment to the figures. Most fluorescent panels use DAPI in all separate channel images, which hinders the actual signal of interest. In my opinion showing DAPI signaling in completely merged images would be enough, and allow for better evaluation of the signals in separated images.

10. The authors mention that they work with Independent ASC isolations from pool of mice and these are considered a biological replicate. In Material&Methods section they mention that these cells are sorted. It is a bit confusing that they mention for example in Figure 1E, F where they show n=5 isolations from 2-4 mice. Are those cell populations coming from the pool of mice, or are those distinct isolations? I think the material and methods paragraph should include clarifying information in that regard.

11. Figure 1D - There are three datapoints on the graphs, but no mention of n in the legend. Is this only one experiment with 3 technical replicates. In material and methods there is information about that all samples were run in biological and technical triplicate.

12. Figure 4B & B' - cilia marker and sensor are not labeled, making the figure difficult to read. B' representative images show that with the Forskolin and FFAR4 agonist the cAMP sensor is weaker at 10minutes, but the graph communicates an accumulation.

13. Figure 5A, B - does not mention what cells are treated and used for this analysis.

Reviewer 3

Advance summary and potential significance to field

In their present study, Lee and Hilgendorf investigated the impact of prostaglandin E2 (PGE2) on adipocyte differentiation (adipogenesis). They found that PGE2 strongly inhibits adipogenesis in both the 3T3-L1 cell line and in primary cells, and that this effect is mediated by the ciliary PGE2 receptor EP4. Surprisingly, they show that this PGE2/EP4 signaling is not mediated by the cAMP/PKA pathway, contrary to what has been previously observed in the context of ciliary EP4 signaling. Instead, they provide compelling evidence that this ciliary EP4 pathway stimulates Rho/ROCK2 activity, promoting cytoplasmic actin fiber formation, which ultimately suppresses adipogenesis. This is a well-conducted and highly interesting study, and the manuscript is well written. However, I have a few minor comments/questions:

1. Adipogenesis should be more clearly defined for readers who are not specialists in adipocyte biology.

2. Rho/ROCK activation is typically associated with negative regulation of ciliogenesis, suggesting that other cilia-dependent signaling pathways might also be affected. Could the authors comment on this? How do they explain the observed increase in cilium length in PGE2-treated cells (Fig. 4)? The results shown in Figure 3 indicate that KIF3A knockout cells are still able to differentiate, suggesting that ciliary signaling is not required for adipogenesis. Based on the introduction, it is unclear whether this result was expected—could the authors clarify?
3. Perhaps I missed it, but it would have been useful to report which PTGERS are expressed in 3T3-L1 cells.
4. Page 6, lines 47-49: "Cilia are lost during adipogenesis and are not present on mature adipocytes (Hilgendorf et al., 2019)." Please also cite earlier studies that support this observation.
5. Page 7, lines 55-58: "...resulting in its accumulation at the centrosome (Fig. S3D, E)." As EP4 is a transmembrane protein, it is unlikely to accumulate at the centrosome, which is a membraneless organelle. Could this be due to nonspecific centriolar staining by the used antibody?
6. Previous studies have shown that stimulation of EP2 and/or EP4 can inhibit Rho/ROCK activity via cAMP/PKA. These findings should be acknowledged and discussed.
7. The authors used 20 μM of PGE2 in their experiments, which is a relatively high concentration considering the strong affinity of EPs for their endogenous ligand. Could they justify this choice?
8. Why did the authors not test EP4-selective agonists?

First revision

Author response to reviewers' comments

We appreciate the opportunity to revise our recent manuscript following the insightful comments we received from all three reviewers. We are grateful for their comments and interest in our manuscript.

Included below is a complete list of each of the reviewers' comments (**bolded**) with a concordant response (**blue**). A version of the paper with highlighted changes has been included in the supplemental information.

Reviewer 1: SUMMARY OF THE ADVANCE MADE IN THIS PAPER AND ITS POTENTIAL SIGNIFICANCE TO THE FIELD

In this study, Lee & Hilgendorf showed that PGE2 inhibit adipogenesis of adipocyte stem cells (ASCs) and 3T3-L1 cells induced by insulin+dexamethasone+IBMX (IDX cocktail). Using specific inhibitors and KO cells, the authors showed that this inhibition is mediated by the EP4/PTGER4 receptor. In addition, using Kif3a-KO cells and Tulp3-KO cells, the EP4-mediated inhibition was shown dependent on cilia and suggested to be dependent on ciliary localization of EP4. Furthermore, the authors showed that the EP4-mediated inhibition is not dependent on canonical cAMP signaling, but, unexpectedly/curiously, dependent on activation of the Rho-ROCK2 pathway, resulting in stabilization of the stress fiber.

Although the mechanism linking EP4 and the Rho-ROCK2 pathway is unclear, I think this paper is worthy of publication in JCS as the findings in this study are novel. However, it is a bit puzzling in light of previous studies involving one of the authors (Hilgendorf KI), and the concerns described below need to be addressed.

SUGGESTIONS TO AUTHORS

Major points:

1. Agonists of FFAR4/GPR120 and those of EP4 were shown to stimulate secretion of insulin and glucagon via cAMP signaling (Wu et al., 2021). In addition, Hilgendorf et al. (2019) showed that omega3-FA stimulates ciliary FFAR4 to induce adipogenesis in 3T3-L1 cells via the cAMP pathway. Although the present study showed that stimulation of 3T3-L1 cells with

FFAR4 agonist but not PGE2 induces cAMP production, it was not investigated what happens to adipogenesis; namely, the question is whether PGE2 inhibits adipogenesis in 3T3-L1 cells induced not only by IDX but also by FFAR4 agonists. In addition, can FFAR4 agonists inhibit stress fiber formation? If so, can PGE2 cancel the FFAR4-mediated inhibition of stress fiber formation? If PGE2 inhibits FFAR4-stimulated adipogenesis, discuss what discriminates between Gs-coupling of FFAR4 and Gs-uncoupling of EP4 in 3T3-L1 cells.

We greatly appreciate the feedback provided by this reviewer. We agree that the interplay between FFAR4 and EP4 signaling in the primary cilium is of significant interest, and are happy to report that we have addressed this point in the revised manuscript.

Specifically, we have included data in *revised Figure 3D* demonstrating that PGE2 inhibits adipogenesis induced by the cilia-dependent FFAR4 agonist cocktail. We further confirm that differentiation induced by FFAR4 agonists also promotes the disassembly of actin stress fibers, and that PGE2 prevents this disassembly (*revised Supplementary Figure 5E*). Finally, we have included additional comments in the *revised Discussion* to address the important question of G protein coupling and ciliary cAMP.

2. The authors showed that ROCK inhibitors (Y-27632 and GSK429286A) antagonize EP4-mediated inhibition of adipogenesis and remodeling of stress fibers. However, the underlying mechanism is unclear. To confirm that these inhibitors do not affect the formation of cilia or the entry of EP4 into cilia, data are needed to verify the rate of cilia formation (cilia length) and the ciliary localization of EP4 upon the inhibitor treatment.

We agree that these are critical controls and we have included new data showing that the ROCK inhibitors do not alter ciliation, EP4 ciliary localization, or cilia length in *revised Supplementary Figure 4G, H*. Gratifyingly, these experiments also revealed that while ROCK inhibition rescues the inhibition of adipogenesis by PGE2/ciliary EP4, it does not rescue or exacerbate the ciliary lengthening induced by PGE2, suggesting that ciliary EP4 activates at least two distinct downstream cellular processes to separately alter cilia length and activate ROCK2 to inhibit adipogenesis. These comments have been added to the *revised Discussion*. We are grateful to the reviewer for suggesting these experiments.

Minor points:

1. Page 7, line 52: Tulp3 is not a motor protein, but is an adaptor protein linking the IFT-A complex and GPCRs (for example, see Mukhopadhyay, et al., *Genes Dev.*, 2010; Badgandi, et al., *JCB*, 2017; Hirano, et al., *MBoC*, 2017).

We apologize for this mistake and have corrected it.

2. Most recently, it was shown that ROCK2 may be involved in ciliogenesis by affecting the transport of ciliated vesicles via actin cytoskeleton remodeling (Smith, et al., *Commun. Med.* 2025). Add this to Discussion.

This citation has been added and addressed within the *revised Results and Discussion*.

Reviewer 2: SUMMARY OF THE ADVANCE MADE IN THIS PAPER AND ITS POTENTIAL SIGNIFICANCE TO THE FIELD

In this manuscript by Lee and Hilgendorf, the authors discover another cell type - the adipocyte stem cells, in which the E-type prostaglandin receptor 4 (EP4) localizes to primary cilia. Within their manuscript, the previously known function of PGE2 and EP4 in adipogenesis is directed by the authors into the ciliary compartment. Herein, the authors are expanding on cell fate mechanisms that are regulated by the primary cilia for fine tuning, and offering insights on how chronic inflammation may result in adipose tissue dysfunction and metabolic disease progression. While the manuscript introduces novelty, I have several comments that need to be addressed in order to further support the statements authors made.

SUGGESTIONS TO AUTHORS

1. P6, l13-17: "PGE2 completely abrogates the differentiation of ASCs isolated from the inguinal white adipose depot of both male and female lean mice (Fig. 1 E, F, FigS1F, G) and similarly inhibits adipogenesis in perigonadal ASCs (Fig S1G-I)" This is not true for the female perigonadal ASCs (Fig. S1I). This appears to correlate with the rather small difference in EP expression of obese vs. lean mice and humans (Fig. S1A, B). The authors should discuss this sex-dependent difference.

We appreciate all of the comments by reviewer 2, which, as addressed, have significantly strengthened our manuscript.

We acknowledge that visceral perigonadal ASC adipogenesis is not significantly decreased by PGE2 addition, but note that these cells at a baseline exhibit lower adipogenesis than their counterparts from males, which makes any further decrease in adipogenesis difficult to assess. We have revised our manuscript to address this and remarking that the difference observed in perigonadal ASCs from female mice may be due to sex-dependent differences in EP expression.

2. A gene knock-out is introduced at several instances in the manuscript (for EP4, KIF3A and TULP4). The authors need to show protein levels of the targeted proteins, such as by using western blot, to make sure about the loss of protein or possible truncations. A related question - if a complete EP4 knock-out was induced, how do you explain EP4+ cilia in these cells (Fig. S2A, B)?

Western blots for both EP4 and KIF3A have been included in *revised Supplementary Figure 1* and *revised Supplementary Figure 3*, respectively. The TULP3 knock-out was previously described (PMID: 31761534), and we've included this reference in the revised manuscript.

To address the second comment, we acknowledge that our EP4 knock-out is not a complete knock out; rather, this cell line is a heterogenous pool of 3T3-L1 cells with variable insertions and deletions within the *Ptger4* gene, as shown in Supplementary Figure 1J. This is commonly done with this cell line, because single cell sorting of 3T3-L1 cells in general (i.e., no KO, just parental) results in dramatic loss of adipogenic capacity. As such, we are not able to generate a completely clonal EP4 knockout. However, our EP4 knockout differentiation data (Figure 1H), TIDE sequencing analysis (Supplementary Figure 1J), EP4 expression data (*revised Supplementary Figure 1L*), and EP4 localization data (Supplementary Figure 2A,B) support our assertion that this knockout has a dramatic decrease in EP4 levels. We have included a comment on the lack of full knock-out within our *revised Results* to reflect this concern.

3. P8, l20-24: "Consistent with previous findings (Dalbay et al, 2015), cilia length transiently increases 24 hours post-initiation of adipogenesis with the IDX cocktail (Fig. 4A, Fig S4A). Addition of PGE2 results in a greater and more persistent increase in cilia length..." The authors do not show any timely dynamics of the cilia length; in the two time points they use, there is no difference in cilia length. Hence, they cannot talk about "transient increase" or "more persistent increase" in cilia length, since they only see elongation that does not go back.

We thank the reviewer for this comment and acknowledge that additional timepoints are necessary to recapitulate the dynamics of cilia length previously reported. Since this manuscript is focused on the effect of PGE2 on cilia length, we have changed the language in the *revised Results* to more accurately represent the data presented.

4. Fig. 3B: Knock-out of TULP3 produces partial inhibition of ciliary EP4 localization, resulting in complete reversal of the inhibitory effect of PGE2 on adipogenesis. This raises a possibility that other receptors and signaling pathways may have been affected, contributing to the loss of adipogenesis phenotype. This needs to be thoroughly discussed in the manuscript.

We appreciate this comment as we agree that TULP3 loss is likely to have a multitude of effects on ASC cilia beyond ciliary EP4. Comments addressing the role of TULP3 trafficking other mediators of adipogenesis to the cilium have been added to the *revised Discussion*.

5. The conclusions on cAMP signaling having negligible-to-no role in the PGE2/EP4-mediated

adipogenesis block seem preliminary. Fig. 4B shows that a single tested concentration of PGE2 produces lesser effect on ciliary cAMP production than a single tested concentration of FFAR4 agonist and forskolin. The effects can, however, be concentration-dependent. The authors need to test at least three concentrations of the respective drugs, in order to show that the PGE2 concentration used throughout the experiment to inhibit adipogenesis indeed produces a relatively minor ciliary cAMP, compared to FFAR4 agonist and forskolin.

We agree that it is possible that higher concentrations of PGE2 would cause a more robust effect on ciliary cAMP. However, we note that the same concentration of PGE2 used in the ciliary cAMP reporter assay is sufficient to inhibit adipogenesis while not generating sustained changes in ciliary cAMP. In contrast, both an agonist of the native cilia-localized G_s-coupled receptor FFAR4 and an adenylyl cyclase agonist were able to induce prolonged increases in ciliary cAMP at the concentrations used in adipogenesis assays. Most importantly, we demonstrate that canonical cAMP effector proteins are not required for PGE2 to inhibit adipogenesis, supporting our conclusion that ciliary EP4 does not inhibit adipogenesis via canonical ciliary cAMP signaling. However, as noted by the reviewer, we cannot exclude the possibility that ciliary EP4 activates some ciliary cAMP, perhaps to modulate cilia length as previously described. Intriguingly, our *revised Supplementary Figure 4* shows that ROCK inhibition rescues PGE2-linked inhibition of adipogenesis but not PGE2-linked increase in cilia length. This is consistent with PGE2 activating several downstream cellular processes, including ones not related to the adipogenesis phenotype. We have expanded our discussion of cAMP and canonical cAMP effector signaling in the *revised Discussion*.

6. Fig. S5A: mRNA levels were quantified in cells treated or not with PGE2 during differentiation. In order to make sure that protein levels also drop in a similar way, the authors need to run ROCK1/2 western blot.

ROCK2 blots have been included in *revised Supplemental Figure 5*.

7. Fig. 5A: The representative images need higher magnification to fully appreciate the phenotype. With the current images, I see no difference between Rho Inh. and Rho Inh.+PGE2 samples.

These changes have been made.

8. Finally, to further support their model (Fig. 5F), the authors need to provide additional experiments targeting the ciliary PEG2/EP4-Rho-A-ROCK2-actin cytoskeleton axis. How is cytoskeleton remodeling (analogous to experiments in Fig. 5C, D) in the EP4-knock-out 3T3-L1 cells? Do you achieve similar results with genetic/epigenetic (CRISPR/siRNA) targeting of ROCK2 (to eliminate the controversy of using small molecule inhibitors)?

We have provided additional data to address these comments and now show in *revised Figure 5G, H* that PGE2 is unable to stabilize stress fibers in the EP4 knock-out cells.

We note that ROCK2 mRNA and protein abundance decrease dramatically during adipogenesis, which complicates genetic/epigenetic approaches to modulate ROCK2 activity during adipogenesis/adipogenesis inhibition. We agree that small molecule inhibitors can have off-target effects (as illustrated in this manuscript), and we have therefore used multiple inhibitors as well as actin staining to support our model that PGE2 inhibits adipogenesis by activating ROCK2 and stabilizing actin stress fibers.

9. A general comment to the figures. Most fluorescent panels use DAPI in all separate channel images, which hinders the actual signal of interest. In my opinion showing DAPI signaling in completely merged images would be enough, and allow for better evaluation of the signals in separated images.

We appreciate the comment and agree that the DAPI can be distracting on some images. DAPI has been removed from the individual channel images for the whole-mount *in vivo* images in *revised*

Figure 2D. However, we do believe that the nuclear marker is helpful to orient readers' eyes in high magnification images, such as the 3T3-L1 and isolated primary ASCs.

10. The authors mention that they work with Independent ASC isolations from pool of mice and these are considered a biological replicate. In Material&Methods section they mention that these cells are sorted. It is a bit confusing that they mention for example in Figure 1E, F where they show n=5 isolations from 2-4 mice. Are those cell populations coming from the pool of mice, or are those distinct isolations? I think the material and methods paragraph should include clarifying information in that regard.

The description in the *revised Materials and Methods* has been clarified. Each biological repeat is an independent isolation that pooled adipose tissue from 2-4 mice prior to sorting to obtain a sufficient number of ASCs for experiments.

11. Figure 1D - There are three datapoints on the graphs, but no mention of n in the legend. Is this only one experiment with 3 technical replicates. In material and methods there is information about that all samples were run in biological and technical triplicate.

The legend has been updated to indicate each data point on the graphs is a biological replicate.

12. Figure 4B & B' - cilia marker and sensor are not labeled, making the figure difficult to read. B' representative images show that with the Forskolin and FFAR4 agonist the cAMP sensor is weaker at 10minutes, but the graph communicates an accumulation.

We have updated *revised Figure 4* with labels for the sensor and constitutive cilia marker. Similarly, we have updated the legend and axis of Figure 4 to better represent its quantification as a ratio of red signal to green signal. As clarified in the *Materials and Methods* section, this is a "down" cAMP sensor, meaning that an accumulation of cAMP reads out as a decrease in cAMP fluorescent intensity (shown in the images), which we then quantified (in the graph).

13. Figure 5A, B - does not mention what cells are treated and used for this analysis.

We apologize for missing this in the figure legend and have updated it to include the cell type.

Reviewer 3: In their present study, Lee and Hilgendorf investigated the impact of prostaglandin E2 (PGE2) on adipocyte differentiation (adipogenesis). They found that PGE2 strongly inhibits adipogenesis in both the 3T3-L1 cell line and in primary cells, and that this effect is mediated by the ciliary PGE2 receptor EP4. Surprisingly, they show that this PGE2/EP4 signaling is not mediated by the cAMP/PKA pathway, contrary to what has been previously observed in the context of ciliary EP4 signaling. Instead, they provide compelling evidence that this ciliary EP4 pathway stimulates Rho/ROCK2 activity, promoting cytoplasmic actin fiber formation, which ultimately suppresses adipogenesis. This is a well-conducted and highly interesting study, and the manuscript is well written. However, I have a few minor comments/questions:

1. Adipogenesis should be more clearly defined for readers who are not specialists in adipocyte biology.

Foremost, we would like to thank reviewer 3 for their comments regarding the quality of our work and manuscript. To address their first comment, we have more clearly defined adipogenesis in our *revised Introduction*.

2. Rho/ROCK activation is typically associated with negative regulation of ciliogenesis, suggesting that other cilia-dependent signaling pathways might also be affected. Could the authors comment on this? How do they explain the observed increase in cilium length in PGE2-treated cells (Fig. 4)? The results shown in Figure 3 indicate that KIF3A knockout cells are still able to differentiate, suggesting that ciliary signaling is not required for adipogenesis. Based on the introduction, it is unclear whether this result was expected—could the authors clarify?

We thank the reviewer for this comment. In addition to demonstrating that PGE2 addition activates

the Rho/ROCK signaling pathway, we have now included data on the changes in ciliation observed with PGE2 treatment in *revised Supplementary Figure 4G, H*. We show that while PGE2 significantly alters cilia length, neither PGE2 nor ROCK alters the overall rate of ciliogenesis. Further, we do not see a decrease in ciliary EP4 localization relative to the non-PGE2 treated control. As to what other ciliary pathways might be affected and how PGE2 treatment/ROCK activation may reshuffle ciliary proteins remains unknown. This has been included in the *revised Discussion*.

Our data strongly suggest ciliary cAMP is not the primary mediator of PGE2 activity on adipogenesis (Figure 4). However, the observation of temporally elevated ciliary cAMP immediately following PGE2 treatment cannot be disregarded altogether. As our data investigating the effects of ROCK inhibition on cilia length demonstrates, ROCK activity is not required for cilia lengthening to occur, indicating there must be an alternative pathway through which PGE2 increases cilia length. We have included in the *revised Results* a potential connection to the ciliary cAMP elevation and cilia length phenotype observed.

The results that KIF3A knock-outs are capable of differentiating is not surprising, as in this manuscript we use the standard differentiation cocktail which contains supraphysiological levels of adipogenesis inducers. As previously shown (PMID: 31761534), cilia are not required for adipogenesis at these very high levels of inducers, but the primary cilium becomes increasingly required as the levels of adipogenesis inducers in the differentiation cocktail are titrated back. We show here that ciliary EP4 inhibits adipogenesis even with the standard differentiation cocktail (very high levels of adipogenesis inducers), but that this requires ciliary localization of EP4. We have included additional data in *revised Figure 3D* demonstrating that PGE2 also inhibits adipogenesis induced by the cilia-dependent FFAR4 agonist cocktail. The *revised Results* have been updated to explain this point more thoroughly.

3. Perhaps I missed it, but it would have been useful to report which PTGERs are expressed in 3T3-L1 cells.

We have included an immunoblot verifying expression of all four EP receptors in 3T3-L1s in *revised Supplementary Figure 1*.

4. Page 6, lines 47-49: "Cilia are lost during adipogenesis and are not present on mature adipocytes (Hilgendorf et al., 2019)." Please also cite earlier studies that support this observation.

These studies have now been cited and included.

5. Page 7, lines 55-58: "...resulting in its accumulation at the centrosome (Fig. S3D, E)." As EP4 is a transmembrane protein, it is unlikely to accumulate at the centrosome, which is a membraneless organelle. Could this be due to nonspecific centriolar staining by the used antibody?

We apologize for this mistake. We believe that EP4 accumulates around the centrosome in vesicles in TULP3 knockout cells. We have revised the language in the *revised Results*.

6. Previous studies have shown that stimulation of EP2 and/or EP4 can inhibit Rho/ROCK activity via cAMP/PKA. These findings should be acknowledged and discussed.

This has been added to the *revised Discussion*.

7. The authors used 20 μ M of PGE2 in their experiments, which is a relatively high concentration considering the strong affinity of EPs for their endogenous ligand. Could they justify this choice?

In Figure 3C, we calculated the EC50 for PGE2 mediated inhibition in 3T3-L1 preadipocytes as 14.28 μ M. As such, we believed 20 μ M to be an easily repeatable and consistent inhibitory concentration, which we importantly show only affects adipogenesis in the presence of cilia, EP4, and ciliary GPCR trafficking. As demonstrated in the dose-response curve, we observe an effect on adipogenesis also at concentrations below 20 μ M.

8. Why did the authors not test EP4-selective agonists?

We appreciate this comment and have added an EP4 specific agonist to *revised Figure 1*.

Second decision letter

MS ID#: jcs.264193R1

MS Title: Prostaglandin E2 inhibits adipogenesis through the cilia-dependent activation of ROCK2

Authors: Mark D. Lee; Keren I Hilgendorf

Article Type: Research Article

Dear Dr Hilgendorf,

I am happy to tell you that your manuscript has been accepted for publication in Journal of Cell Science, pending standard publication integrity checks.